# The nucleic acid binding protein SFPQ represses EBV lytic reactivation by promoting histone H1 expression

Laura A. Murray-Nerger[1,2,3,4], Clarisel Lozano[1], Eric M. Burton[1,2,3,4], Yifei Liao[1,2,3,4], Nathan A. Ungerleider[5], Rui Guo [6] & Benjamin E. Gewurz [1,2,3,4] ✉

Epstein-Barr virus (EBV) uses a biphasic lifecycle of latency and lytic reactivation to infect >95% of adults worldwide. Despite its central role in EBV persistence and oncogenesis, much remains unknown about how EBV latency is maintained. We used a human genome-wide CRISPR/Cas9 screen to identify that the nuclear protein SFPQ was critical for latency. SFPQ supported expression of linker histone H1, which stabilizes nucleosomes and regulates nuclear architecture, but has not been previously implicated in EBV gene regulation. H1 occupied latent EBV genomes, including the immediate early gene *BZLF1* promoter. Upon reactivation, SFPQ was sequestered into subnuclear puncta, and EBV genomic H1 occupancy diminished. Enforced H1 expression blocked EBV reactivation upon SFPQ knockout, confirming it as necessary downstream of SFPQ. SFPQ knockout triggered reactivation of EBV in B and epithelial cells, as well as of Kaposi's sarcoma-associated herpesvirus in B cells, suggesting a conserved gamma-herpesvirus role. These findings highlight SFPQ as a major regulator of H1 expression and EBV latency.

Epstein-Barr virus (EBV) is a γ-herpesvirus that persistently infects over 95% of human adults worldwide. EBV contributes to multiple B-cell lymphomas and to gastric and nasopharyngeal carcinomas[1-4]. Together, these comprise >200,000 cancer cases per year. EBV causes infectious mononucleosis and is a multiple sclerosis trigger[5]. To accomplish lifelong infection, EBV uses a biphasic lifecycle that relies on colonization of the long-lived memory B-cell reservoir. While environmental triggers can induce reactivation from latency, including memory cell differentiation into plasma cells, mechanisms that regulate the viral lytic switch remain incompletely understood. Elucidating these interactions is of particular interest because such mechanisms may be leveraged for lytic induction therapeutic strategies to target cancer cells that harbor latent EBV genomes[6].

Maintenance of EBV latency relies on multiple layers of viral genome epigenetic regulation[1,7-10]. Upon primary infection, EBV genomes enter the nucleus as unchromatinized, linear double-stranded DNA. However, they are rapidly chromatinized and circularized by host enzymatic machinery to silence the majority of EBV genes as the virus enters a latent state. During latent infection, the EBV genome is replicated by host cell machinery along with the host genome in S-phase, thus ensuring that the EBV genome is distributed to the daughter cells[11]. Upon lytic reactivation, a temporal cascade of viral gene expression is initiated[12]. This cascade relies on the production of the immediate early proteins BZLF1 and BRLF1, which drive expression of 40 viral early genes. Production of early proteins drives lytic replication of EBV genomes, which enables expression of 36 late gene products, packaging of the viral genomes into capsids, and release of infectious virus[12].

A key aspect of the EBV lifecycle that remains poorly understood is the suite of host factors that maintain the EBV genome in a quiescent

[1]Division of Infectious Diseases, Department of Medicine, Brigham and Women's Hospital, 181 Longwood Avenue, Boston, MA 02115, USA. [2]Department of Microbiology, Harvard Medical School, Boston, MA 02115, USA. [3]Harvard Program in Virology, Boston, MA 02115, USA. [4]Center for Integrated Solutions to Infectious Diseases, Broad Institute of Harvard and MIT, Cambridge, MA 02142, USA. [5]Department of Pathology, Tulane University, New Orleans, LA 70118, USA. [6]Department of Molecular Biology and Microbiology, Tufts University, Medford, MA 02155, USA. ✉e-mail: bgewurz@bwh.harvard.edu

state. DNA methylation functions in regulating EBV latency and lytic reactivation[8,13–15]. Histones add a key additional layer of genomic regulation critical for the maintenance of herpesvirus latency. For instance, the histone H3 loaders CAF1 and ATRX each have important roles in suppression of lytic reactivation[16,17]. However, potential roles for linker H1 histones, which bind nucleosomal core particle DNA entry and exit sites and have major roles in nucleosome stabilization and higher-order chromatin architecture and compaction[18,19], remain to be characterized in the context of herpesvirus latency.

Host transcription factors further regulate EBV latency, either through direct action on viral gene expression or via regulation of host factors important for latency[20–22]. An intriguing class of DNA-binding proteins are the *Drosophila* behavior/human splicing (DBHS) proteins. These proteins are obligate dimers[23] whose nucleic acid binding capabilities allow them to function as a molecular bridge between nuclear DNA and RNA processes. The DBHS family member splicing factor proline and glutamine rich (SFPQ, also known as PSF) has been established as a versatile regulator of transcription[24–28] and of RNA processing[23,29,30]. SFPQ functions as part of a dimeric protein complex that relies on its coiled-coil domain, and it can form either homo or heterodimers depending on the functional context[23]. However, nearly all virus-based studies have focused on SFPQ in the context of RNA virus infection[31–38]. Despite its multifunctional roles in nucleic acid biology, the roles of SFPQ in EBV latency remain unstudied.

Here, we use systematic CRISPR analyses to define a major role for SFPQ in regulation of the EBV lytic switch across B and epithelial cell environments. We demonstrate a role for SFPQ in suppression of Kaposi's sarcoma-associated virus (KSHV) reactivation, suggesting a conserved γ-herpesvirus role. Mechanistically, we identify that SFPQ represses EBV lytic reactivation by supporting linker histone H1 expression and show that enforced H1 expression prevents EBV lytic reactivation upon SFPQ knockout. H1 occupancy at key EBV lytic gene promoters is strongly reduced upon EBV reactivation, further suggesting H1 pro-latency roles. EBV induces redistribution of SFPQ within the nucleus at early stages of lytic reactivation. Together, these results implicate SFPQ and histone H1 as major host factors that maintain highly restricted forms of EBV latency.

## Results

### SFPQ represses lytic reactivation of EBV

To identify host factors critical for the maintenance of EBV B-cell latency, we recently performed a human genome-wide CRISPR/Cas9 screen in Burkitt lymphoma P3HR-1 B-cells[21]. Systematic CRISPR loss-of-function analysis identified SFPQ as a top hit, in which all four single guide RNAs (sgRNAs) targeting SFPQ were highly enriched among cells triggered for lytic reactivation[21] (Fig. 1a, b). Despite its strong CRISPR signal, potential roles for SFPQ in EBV latency maintenance have remained uncharacterized. Given the pleotropic roles of SFPQ at the transcriptional and post-transcriptional levels (Fig. 1c), we hypothesized that SFPQ may suppress EBV lytic reactivation through effects on the EBV genome.

To validate that SFPQ is important for EBV latency, we knocked out (KO) SFPQ by expressing either of two independent sgRNAs in three Cas9+ Burkitt lymphoma cell lines: P3HR-1, MUTU I, and Daudi. In all three cell lines, SFPQ KO robustly induced the EBV immediate early BZLF1 and the early BMRF1 proteins (Fig. 1d). Moreover, SFPQ triggered a productive lytic cycle, as evidenced by significantly increased plasma membrane expression of the EBV late protein gp350 (Fig. 1e and Supplementary Fig. 1b–e) and increased intracellular and extracellular EBV genome copy numbers (Fig. 1f and Supplementary Fig. 1a). Notably, since MUTU I and Daudi cells are latently infected with type I EBV, while P3HR-1 cells are latently infected with type II EBV, these data indicate a conserved role for SFPQ in maintaining EBV latency across the two major EBV types. SFPQ KO also reactivated EBV lytic gene expression in both YCCEL1 and SNU-719 gastric carcinoma and in

C666.1 nasopharyngeal carcinoma cell lines (Fig. 1d). Therefore, SFPQ is important for maintaining EBV latency in both B and epithelial cells.

We next used cDNA rescue to validate that on-target CRISPR editing effects on *SFPQ* were responsible for EBV lytic reactivation. To do so, we engineered a silent point mutation into the *SFPQ* cDNA protospacer-adjacent motif (PAM) site targeted by SFPQ sgRNA #1 to prevent Cas9-mediated cutting and subsequent editing. We generated MUTU I cells with stable expression of either HA-epitope tagged GFP (HA-GFP) as a negative control or the HA-tagged SFPQ rescue cDNA (HA-SFPQ[R]). HA-SFPQ[R] properly localized to the nucleus and exhibited a similar subnuclear distribution pattern as endogenous SFPQ (Fig. 1g). Upon KO of endogenous SFPQ, expression of the HA-SFPQ[R], but not control HA-GFP, repressed lytic reactivation (Fig. 1h), confirming an obligatory SFPQ role in maintenance of EBV latency.

Given the pro-latency role of SFPQ across multiple cell types, we hypothesized that SFPQ may broadly regulate γ-herpesvirus latency. Therefore, we knocked out SFPQ in the primary effusion lymphoma (PEL) cell line JSC-1, which is latently co-infected by both EBV and the γ-herpesvirus KSHV. SFPQ depletion induced expression of the KSHV early lytic protein ORF57, indicative of KSHV reactivation, as well as of BZLF1 and BMRF1 (Supplementary Fig. 1f). To then determine if EBV and KSHV reactivated within the same PEL cells, we performed immunofluorescence microscopy. Within a given cell, SFPQ KO reactivated either EBV or KSHV, but not both viruses. This observation raises the possibility that EBV and KSHV may have ways to repress each other's lytic cycles (Supplementary Fig. 1g–i).

### SFPQ does not require MYC or NONO to repress lytic reactivation

We recently identified MYC, along with a series of proteins that control MYC expression, as critical for EBV latency[21]. Given that SFPQ functions in transcriptional and post-transcriptional regulation, we next tested whether SFPQ indirectly controls EBV latency via effects at the level of MYC. Cas9+ MUTU I cells with stable lentivirus-driven GFP control or MYC cDNAs were established. We then expressed control or SFPQ sgRNAs. Notably, in contrast to all hits examined in our original CRISPR screen analysis[21], MYC over-expression could not rescue EBV lytic reactivation upon SFPQ KO, as evidenced by similar levels of BZLF1, BMRF1, and gp350 protein levels and by similar increases in EBV genome copy number in control GFP and MYC overexpressing cells (Fig. 2a–c and Supplementary Fig. 2a, b). These data suggest that SFPQ and MYC control EBV latency by distinct pathways, or that SFPQ may instead function downstream of MYC to maintain EBV latency.

We next asked whether SFPQ KO might deregulate EBV lytic gene expression by simultaneously de-repressing immediate early, early, and late genes. There is precedence for such a hypothesis in herpes simplex virus type 1 (HSV-1) lytic reactivation where a burst of viral gene expression can be observed that does not rely on viral immediate early genes[39–41]. Therefore, to test if SFPQ KO triggers EBV reactivation in a manner dependent on immediate early BZLF1 production, we employed CRISPR to generate *BZLF1* KO MUTU I cells. As a control, we also generated MUTU I cells knocked out for the EBV *BXLF1* gene, which is not required for lytic replication (Supplementary Fig. 2c)[42]. Upon SFPQ KO, EBV early BMRF1 and late gp350 protein expression was observed in *BXLF1* KO control cells, but not in *BZLF1* KO cells (Fig. 2d, e and Supplementary Fig. 2d). These data support a model in which SFPQ represses EBV lytic reactivation at the level of EBV immediate early gene expression, but do not exclude potential additional downstream regulatory roles in repression of early or late genes.

SFPQ can either function independently as a homodimer[34] or together with non-POU domain-containing octamer-binding protein (NONO) as a heterodimer[23,43]. To ascertain whether NONO is required for maintenance of EBV latency, we established control or *NONO* KO P3HR-1 cells. In contrast to SFPQ KO, NONO KO was not sufficient to induce EBV lytic reactivation, which suggests that SFPQ does not

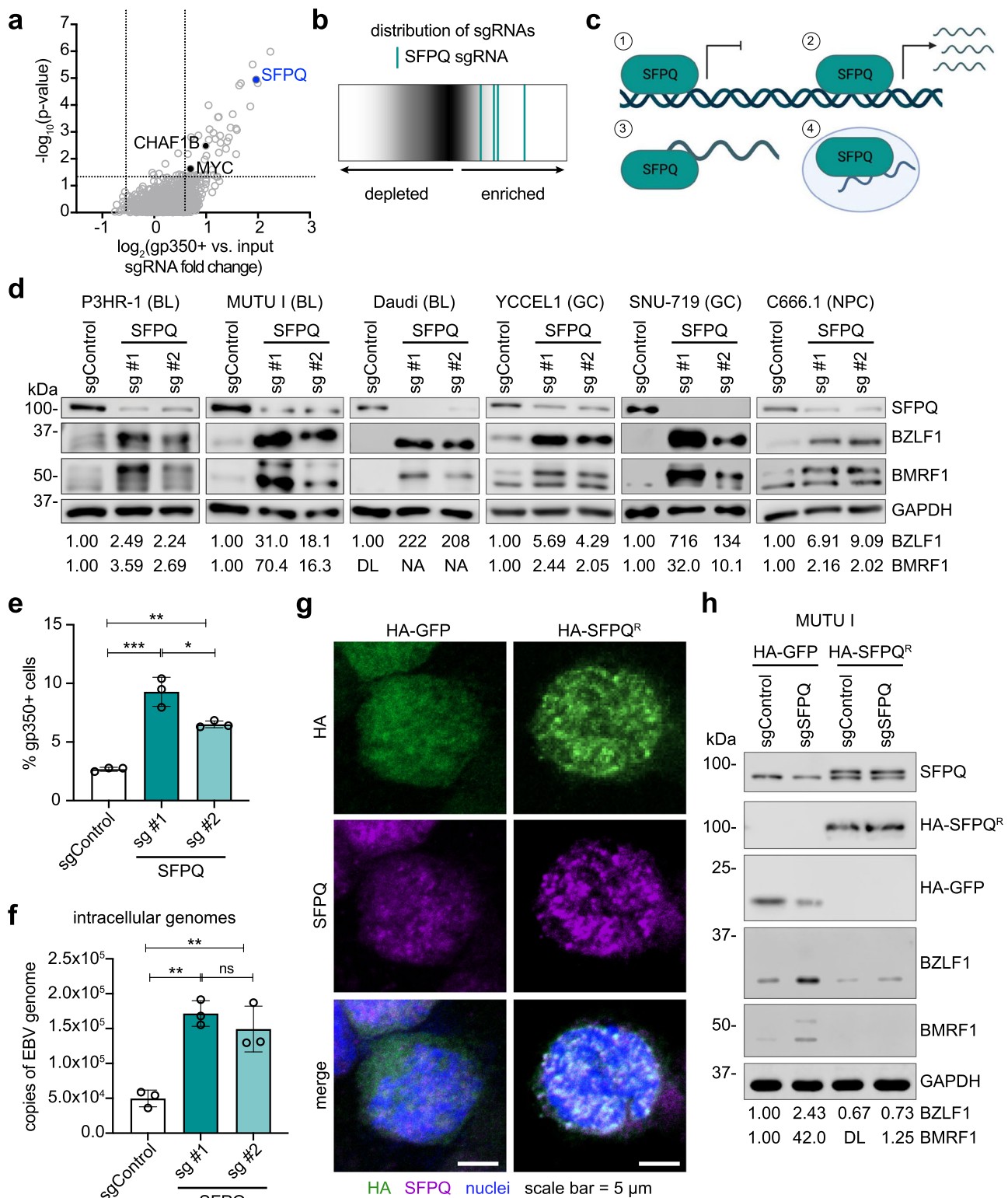

**g** HA-GFP / HA-SFPQ^R

HA SFPQ nuclei   scale bar = 5 μm

require NONO to maintain EBV latency (Fig. 2f). Collectively, these data support a model in which SFPQ represses EBV lytic reactivation independently from MYC or NONO (Fig. 2g).

**SFPQ KO suppresses histone H1 and induces interferon stimulated gene expression**

To gain further insights into how SFPQ supports EBV latency, we performed RNA sequencing (RNA-seq) of Cas9+ Burkitt lymphoma cells at day 6 post expression of control or either of two

independent SFPQ sgRNAs (Fig. 3a, b and Supplementary Data 1). This was the same timepoint at which these sgRNAs strongly scored in the CRISPR screen[21]. Since SFPQ can alter gene expression at the level of transcription or RNA stability, we employed rRNA depletion, rather than polyA enrichment, so that we could broadly capture SFPQ KO effects. SFPQ KO by either sgRNA significantly induced expression of EBV lytic genes spanning immediate early, early, and late genes (Supplementary Fig. 3d and Supplementary Data 1).

**Fig. 1 | SFPQ represses EBV lytic reactivation. a** Volcano plot of −log$_{10}$($p$ value) vs. log$_2$(gp350+ vs. input sgRNA fold change) on Day 6 post lentiviral transduction of P3HR-1 cells (data from ref. 21). Previously characterized regulators of lytic reactivation are highlighted in black. SFPQ is labeled in blue. Data represent three biological replicates. **b** Log$_2$(fold change) of the four guides targeting SFPQ (teal) compared to the distribution of all sgRNA guides from the CRISPR screen (data from ref. 21). **c** Schematic depicting the four main functions known for SFPQ: (1) transcriptional repression, (2) transcriptional activation, (3) mRNA binding, and (4) co-localization with *NEAT1* lncRNA in paraspeckles. **d** Immunoblot analysis of EBV BZLF1 and BMRF1 from whole cell lysates (WCL) obtained from three Burkitt lymphoma (BL), two gastric carcinoma (GC), and one nasopharyngeal carcinoma (NPC) Cas9+ cell lines expressing control or SFPQ sgRNAs. **e** Mean ± standard deviation of % gp350+ Cas9+ P3HR-1 cells from $n = 3$ biological replicates following expression of control or SFPQ sgRNAs. **f** Mean ± standard deviation of intracellular EBV genome copy number from $n = 3$ biological replicates of Cas9+ P3HR-1 cells expressing control or SFPQ sgRNAs. **g** Representative immunofluorescence microscopy images of HA-GFP (control) or CRISPR-resistant HA-SFPQ$^R$ expression (green) with co-staining for endogenous SFPQ (magenta) and nuclear DAPI (blue) in Cas9+ MUTU I cells. Scale bar = 5 μm. Images are representative of $n = 3$ biological replicates. **h** Immunoblot analysis of WCL from Cas9+ MUTU I cells expressing control GFP or CRISPR-resistant SFPQ cDNA (HA-SFPQ$^R$) together with control or endogenous SFPQ targeting sgRNAs. Immunoblots are representative of $n = 3$ biological replicates and densitometry values normalized to the loading control GAPDH are shown. DL indicates below the detection limit. NA indicates not applicable because the control value is below the detection limit. *$p \leq 0.05$, **$p \leq 0.01$, ***$p \leq 0.001$, ns not significant were calculated by one-way ANOVA. Source data are provided as a Source Data file.

At a fold change >2 and multiple hypothesis test adjusted $p$ value <0.05 cutoff, 757 and 761 host genes were upregulated and downregulated by SFPQ sgRNA #1, respectively. Similarly, 251 and 180 host genes were upregulated and downregulated by SFPQ sgRNA #2, respectively (Fig. 3b and Supplementary Fig. 3a–c). Notably, sgRNA #1 more strongly depleted SFPQ expression, potentially accounting for the larger number of differentially expressed genes (Supplementary Fig. 3a–d). Volcano plot analysis highlighted that histone genes were among the most highly suppressed by SFPQ KO (Fig. 3b). The genes encoding the histone H1 variants H1.2 (*HIST1H1C*, also called histone H1C), H1.4 (*HIST1H1E*, also called histone H1E), and H1.5 (*HIST1H1B*, also called histone H1B) were among the most significantly de-repressed (Fig. 3b). SFPQ KO also decreased expression of genes encoding core histones H2A, H2B, H3, and H4, but to a somewhat lesser extent than genes encoding H1 (Fig. 3b). Indeed, genes decreased by SFPQ KO were the most highly enriched for chromatin assembly and organization by Gene Ontology (GO) biological process analysis (Fig. 3c and Supplementary Fig. 3e). While GO terms related to cell cycle progression were only enriched in the SFPQ sgRNA #1 dataset (Fig. 3c and Supplementary Fig. 3e), we assessed the role of SFPQ in cell cycle progression. SFPQ KO did not significantly alter the accumulation of cells in G2 phase (Supplementary Fig. 4a–f). While we observed a modest decrease of cells in S phase upon SFPQ KO, we observed a similar decrease in S phase cell percentage in MUTU I cells upon lytic reactivation via α-IgM cross-linking (Supplementary Fig. 4b–f). These findings align with the known phenomenon that EBV lytic reactivation induces cell cycle arrest in G1 phase[44,45]. These results are consistent with a model in which the observed changes in cell cycle related gene expression upon SFPQ KO are due to EBV lytic reactivation.

GO analyses also highlighted cytokine-mediated signaling and antiviral defense pathways as the most highly induced pathways by SFPQ KO (Fig. 3d and Supplementary Fig. 3e). Multiple interferon stimulated genes (ISGs) were among the most significantly upregulated by SFPQ KO, including *SAMD9* and *IFI44L* (Fig. 3b). Similarly, two of the lncRNAs most highly upregulated by SFPQ KO are associated with ISG expression (*HCP5* and *THRIL*)[46,47] (Supplementary Fig. 3f). Increased expression of the ISGs IFIT1 and IRF7 was validated upon SFPQ KO at the protein level in both P3HR-1 (IFIT1 and IRF7) and MUTU I (IFIT1) cells (Supplementary Fig. 3g, h). ISG expression was not a response to EBV lytic reactivation, as SFPQ KO similarly upregulated IFIT1 in an EBV-negative MUTU I subclone[48] (Supplementary Fig. 3h). We speculate that altered histone H1 expression may underlie this phenotype.

**SFPQ driven linker histone H1 expression is critical for EBV latency maintenance**

Linker histone H1 maintains nucleosome compaction and can thereby repress transcription[49,50] (Fig. 4a). Taken together with our RNA-seq results, we hypothesized that SFPQ supports histone H1 levels to promote EBV latency. Consistent with this, SFPQ depletion reduced H1.2 and H1.4 at the mRNA and protein levels in P3HR-1 and MUTU I

cells (Fig. 4b, c and Supplementary Fig. 5a, b). To determine whether EBV reactivation was required for the effects of SFPQ KO on H1 expression, we expressed control or SFPQ sgRNAs in a Cas9+ EBV negative (EBV−) MUTU I subclone. However, we again observed that SFPQ KO reduced H1.2 and H1.4 mRNA and protein levels (Fig. 4d, e). Demonstrating a key SFPQ role in support of H1 expression even in cells that had never been EBV-infected, SFPQ KO in Cas9+ Ramos B-cells and h-TERT immortalized normal oral keratinocytes (NOK) also reduced H1.2 and H1.4 protein levels (Supplementary Fig. 5c, d). These data indicate that SFPQ supports H1 expression, likely through effects on H1 transcription or RNA stability.

Since SFPQ can regulate gene expression at the DNA or RNA levels[23–30], we next combined CRISPR KO with cDNA rescue to test the roles of the SFPQ DNA binding domain (DBD) and RNA-recognition motifs (RRM) 1 or 2 in maintenance of EBV latency. We generated a panel of MUTU I cells with stable expression of CRISPR-resistant full length SFPQ or deletion mutant cDNAs encoding SFPQ lacking the DBD (ΔDBD), RRM1 (ΔRRM1), or RRM2 (ΔRRM2) domains (Fig. 4f and Supplementary Fig. 5e). We confirmed via confocal microscopy that all four SFPQ deletion mutants exhibited similar subnuclear distribution, which was distinct from that of control GFP (Supplementary Fig. 5f). We then knocked out endogenous SFPQ and assessed whether any of the deletion mutants could maintain EBV latency. As expected, lytic BZLF1 and BMRF1 protein expression was induced by SFPQ KO in cells with negative control GFP expression. Full length, ΔRRM1, or ΔRRM2 SFPQ repressed EBV lytic reactivation to a similar extent, as judged by immunoblot for BZLF1 and BMRF1. In contrast, EBV reactivated to a similar extent in cells expressing ΔDBD SFPQ as in cells expressing the GFP negative control (Fig. 4g).

Based on these findings, we assessed whether full length or ΔDBD SFPQ associates with the *H1.2* promoter by expressing cDNAs expressing HA-tagged versions of full length or ΔDBD SFPQ and performing an α-HA-chromatin immunoprecipitation (ChIP) followed by qPCR. Full length, but not ΔDBD, SFPQ associated with the *H1.2* promoter (Fig. 4h), as well as with the *HEXIM1* promoter (Supplementary Fig. 5g), which was previously reported as a human SFPQ target[51]. These data support a model in which SFPQ DNA binding promotes H1 transcription in support of EBV latency. We also used these same constructs to investigate whether SFPQ associated with the EBV genome. Indeed, ChIP-qPCR revealed that SFPQ also associates with the *BZLF1* promoter, *oriLyt*$^L$ and *oriLyt*$^R$ enhancers, and the C promoter (Cp), again in a DBD-dependent manner (Supplementary Fig. 5h–k). Because these sites include regions important for latency vs. lytic cycle gene expression, SFPQ may broadly associate with the EBV genome in addition to regulating human gene expression.

**H1 is required for EBV latency and occupies key viral genomic elements**

To next test whether H1 was necessary for EBV latency maintenance, we used CRISPR to deplete H1.2, H1.4, or both, as they were the H1

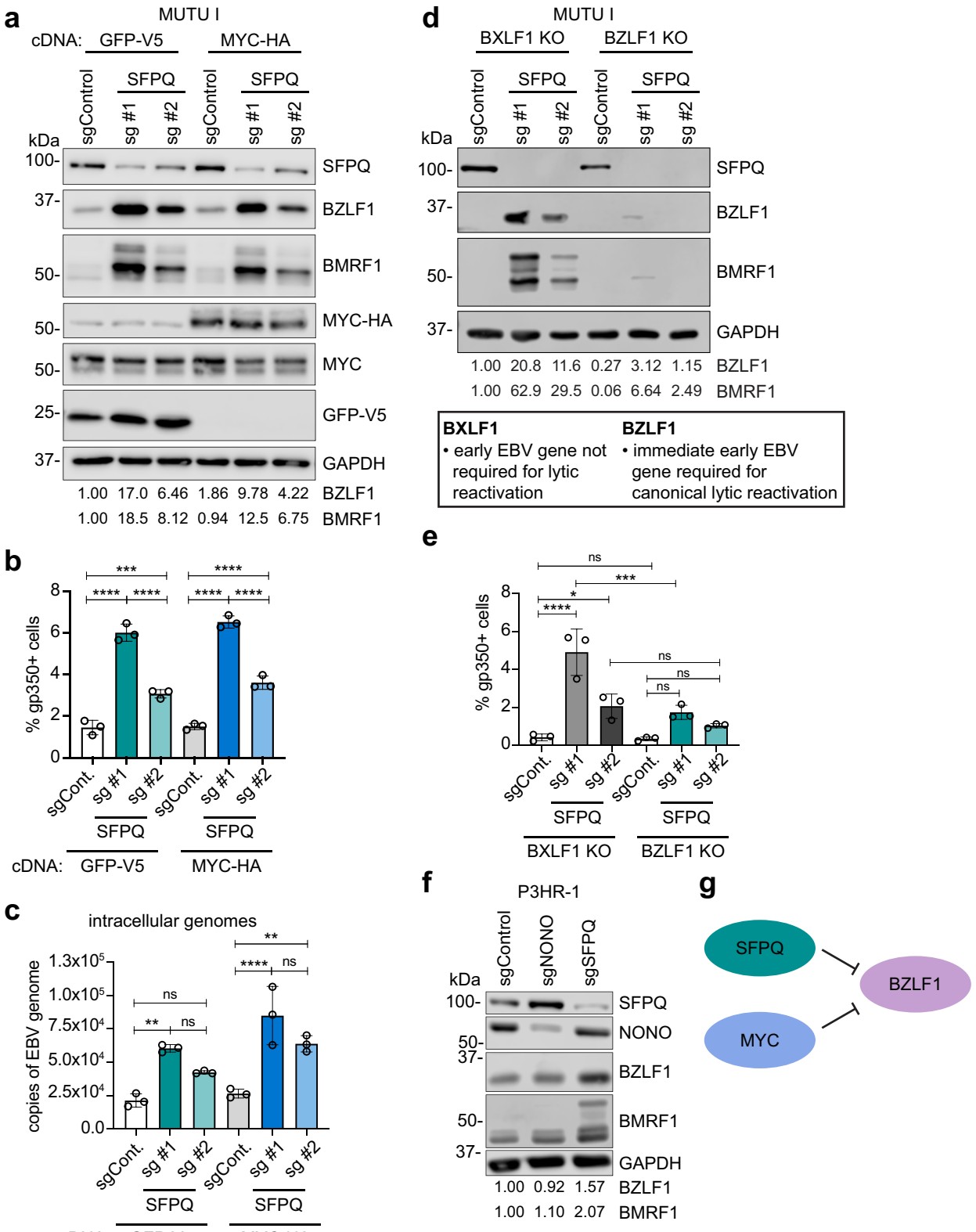

variants most highly expressed in P3HR-1 cells. Depletion of either H1.2 or H1.4 alone only modestly increased BZLF1 and BMRF1 protein expression. However, consistent with redundancy between histone 1 variants[52], depletion of both H1.2 and H1.4 more strongly induced BZLF1 and BMRF1 (Fig. 5a and Supplementary Fig. 6a). Notably, H1 CRISPR targeting did not perturb histone expression more generally,

since similar levels of histone H3 were observed across all depletion conditions (Fig. 5a and Supplementary Fig. 6a).

Given that H1 depletion is sufficient for EBV reactivation, we next tested whether loss of H1 is necessary for SFPQ KO effects on EBV lytic gene expression. We established Cas9+ MUTU I cells with stably expressed V5-epitope tagged control GFP, H1.2, H1.4 or, as a further

**Fig. 2 | SFPQ represses early stages of EBV lytic reactivation in a MYC and NONO independent manner. a** Immunoblot analysis of BZLF1 and BMRF1 using WCL from Cas9+ MUTU I cells expressing control GFP-V5 or MYC-HA cDNA and also control or SFPQ sgRNAs. **b** Mean ± standard deviation % gp350+ cells from $n = 3$ biological replicates of Cas9+ MUTU I cells expressing GFP-V5 or MYC-HA cDNA upon expression of either control or SFPQ sgRNAs. **c** Mean ± standard deviation of intracellular EBV genome copy number from $n = 3$ biological replicates of Cas9+ MUTU I cells expressing GFP-V5 or MYC-HA cDNA and also control or SFPQ sgRNAs. **d** Immunoblot analysis of WCL from Cas9+ MUTU I cells expressing sgRNAs targeting EBV early gene BXLF1 (which is not required for lytic replication) or

immediate early gene BZLF1, as well as control or SFPQ targeting sgRNAs. **e** Mean ± standard deviation % gp350+ cells from $n = 3$ biological replicates of Cas9+ MUTU I cells expressing BXLF1 or BZLF1 sgRNAs, as well as control or SFPQ sgRNAs. **f** Immunoblot analysis of WCL from Cas9+ P3HR-1 cells expressing control, NONO, or SFPQ sgRNAs. **g** Schematic indicating that MYC and SFPQ operate in parallel pathways to repress EBV lytic reactivation. Immunoblots are representative of $n = 3$ biological replicates and densitometry values normalized to the loading control GAPDH are shown. *$p \leq 0.05$, **$p \leq 0.01$, ***$p \leq 0.001$, ****$p \leq 0.0001$, ns not significant were calculated by one-way ANOVA. Source data are provided as a Source Data file.

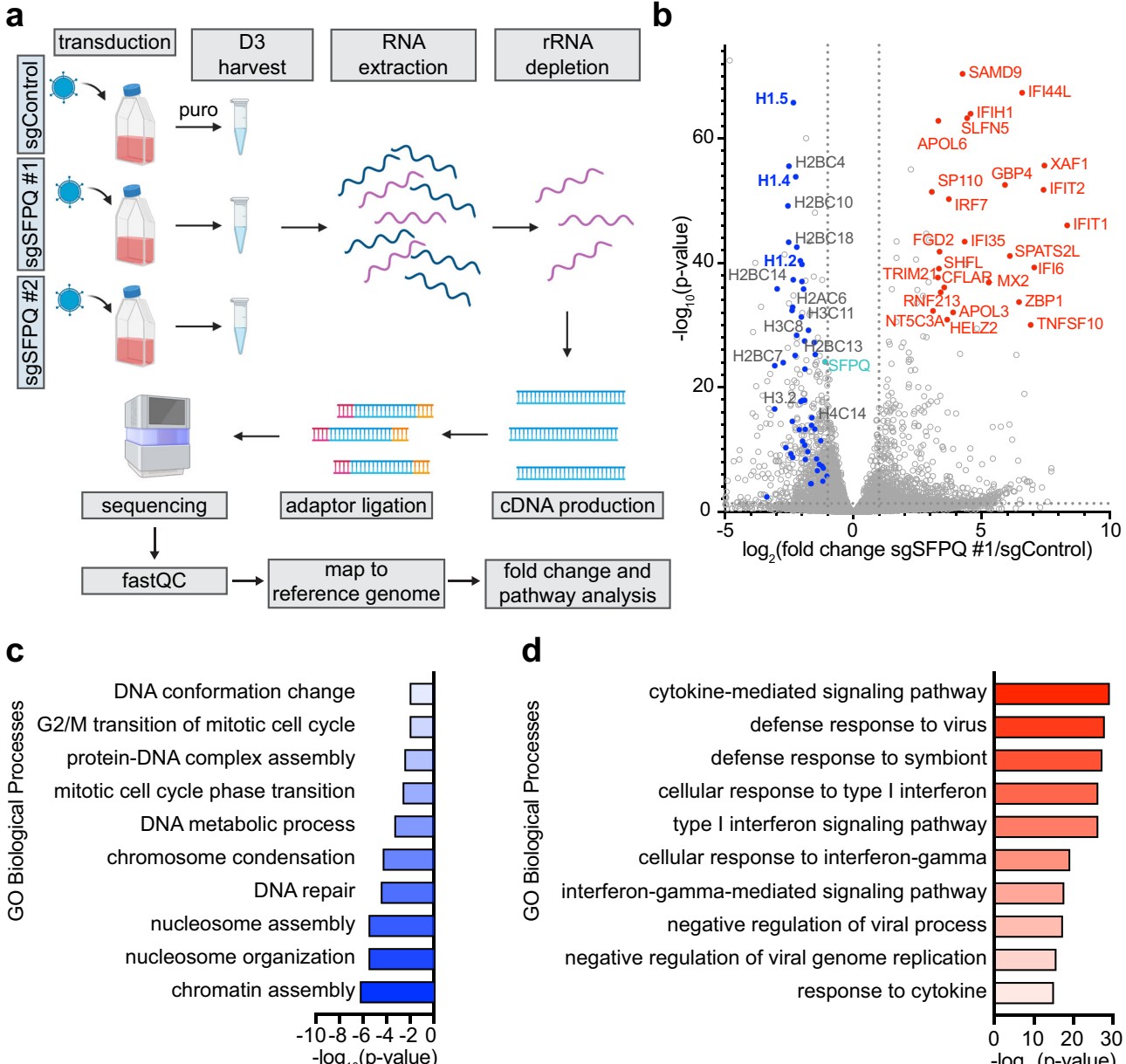

**Fig. 3 | SFPQ regulates expression of chromatin organization and interferon-related genes. a** Workflow of the RNA-seq analysis performed. Briefly, SFPQ was knocked out in Cas9+ P3HR-1 cells via transduction using lentiviruses that expressed control or either of two independent SFPQ sgRNAs. Transduced cells were puromycin (puro) selected. On Day 3 post selection (Day 6 post transduction), cells were harvested. RNA-seq was performed on ribosomal RNA (rRNA) depleted RNA. **b** Volcano plot of the −log$_{10}$($p$ value) vs. log$_2$(fold change) of mRNA

expression in cells expressing SFPQ sgRNA #1 vs. control sgRNA. The adjusted $p$ value (corrected for multiple testing via the Benjamini and Hochberg method) was used. Selected genes that were differentially expressed are highlighted in red (increased) or blue (decreased). **c, d** The top 10 significantly enriched (adjusted $p$ value < 0.05 calculated using the Benjamini and Hochberg method) gene ontology (GO) Biological Processes of genes that were (**c**) decreased or (**d**) increased in abundance in SFPQ depleted (sg #1) vs. control cells.

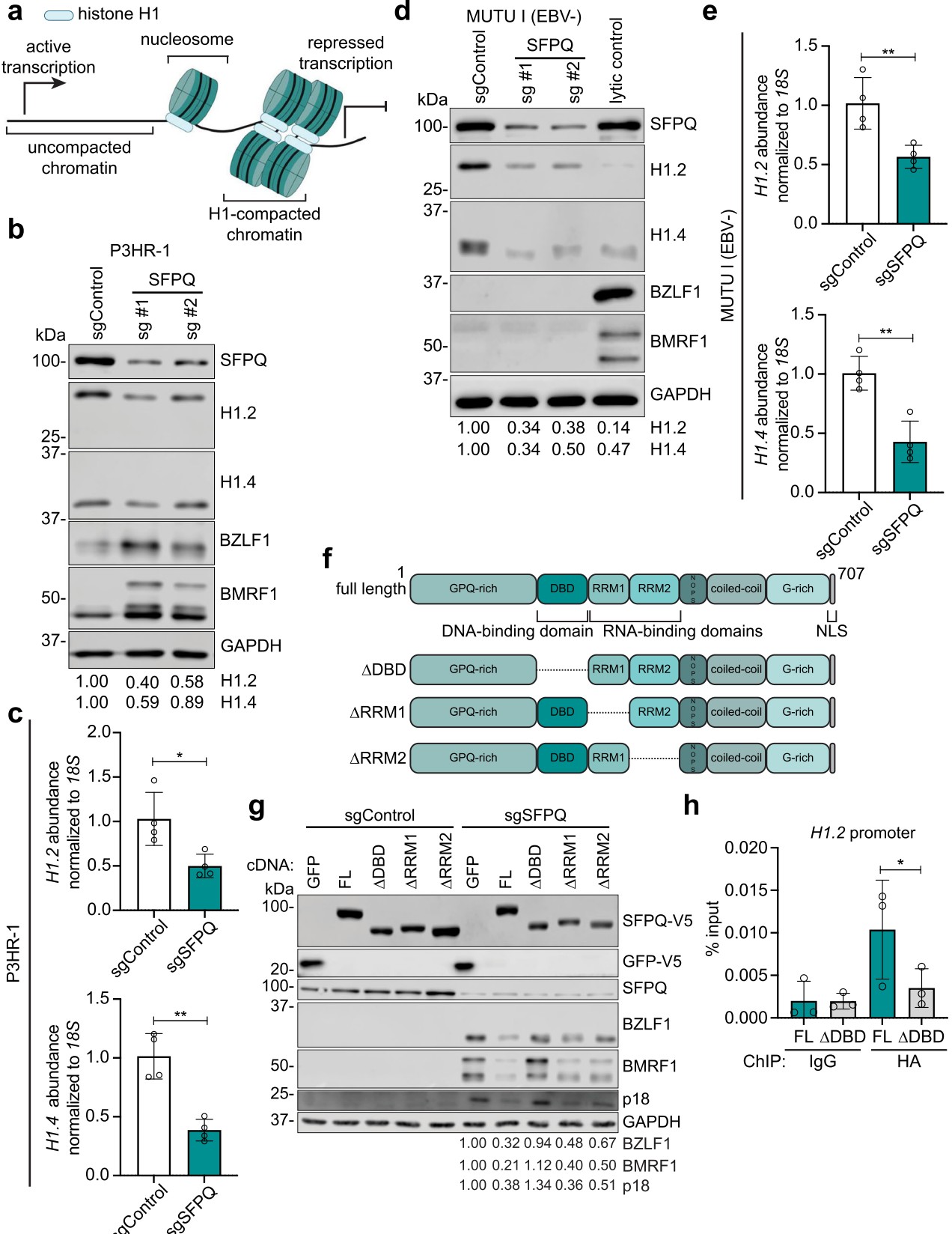

control, H3 (encoded by *HIST1H3B*). We then expressed control or SFPQ targeting sgRNAs and monitored effects on EBV lytic gene expression. Consistent with our model, enforced H1.2 or H1.4, but not H3 expression impeded EBV reactivation upon SFPQ KO (Fig. 5b). Furthermore, enforced expression of the transcription repressor C-terminal-binding protein 1 (CTBP1) also failed to impede EBV reactivation upon SFPQ KO (Supplementary Fig. 6b). Together, these data suggest that SFPQ inhibits EBV lytic reactivation by specifically promoting the expression of linker histone H1, especially the H1.2 and H1.4 variants.

**Fig. 4 | SFPQ regulates expression of linker histone H1. a** Schematic illustrating linker histone H1 (blue oval) binding to inter-nucleosomal DNA to drive chromatin compaction and repress transcription. **b** Immunoblot analysis of H1.2 and H1.4 protein levels in WCL of Cas9+ P3HR-1 cells expressing control or SFPQ sgRNAs. **c** RT-qPCR analysis of mean ± standard deviation of *18S* RNA normalized *HIST1H1C (H1.2)* and *HIST1H1E (H1.4)* abundances from *n* = 4 biological replicates of Cas9+ P3HR-1 cells expressing control or SFPQ sgRNAs. *P* values were calculated using a two-tailed Student's *t* test. **d** Immunoblot analysis of H1.2 and H1.4 protein levels in WCL of Cas9+ EBV− MUTU I cells expressing control or SFPQ sgRNAs. **e** RT-qPCR analysis of mean ± standard deviation of *18S* RNA normalized *HIST1H1C (H1.2)* and *HIST1H1E (H1.4)* abundances from *n* = 4 biological replicates of Cas9+ EBV− MUTU I cells expressing control or SFPQ sgRNAs. *P* values were calculated using a two-tailed Student's *t*-test. **f** Schematic of full length vs. domain deletion SFPQ constructs that were generated. **g** Immunoblot analysis of EBV lytic proteins in WCL from Cas9+ EBV+ MUTU I cells that expressed the indicated SFPQ cDNA construct refractory to CRISPR editing, along with control or SFPQ sgRNAs. **h** ChIP-qPCR analysis of HA-FL or HA-ΔDBD occupancy of the *H1.2* promoter in Cas9+ MUTU I cells. Mean ± standard deviation from *n* = 3 biological replicates is shown with *p* value calculated by one-way ANOVA. Representative immunoblots from *n* = 3 biological replicates and densitometry quantification with values normalized to the loading control GAPDH are shown. *$p \leq 0.05$, **$p \leq 0.01$. Source data are provided as a Source Data file.

H1 functions to promote chromatin compaction to impede gene expression[49,50], but H1 roles in γ-herpesvirus latency were understudied. Therefore, we used ChIP-qPCR to investigate H1 occupancy of the EBV genome in latency and upon lytic reactivation. We focused on the *BZLF1* promoter and the two EBV origins of lytic replication (*oriLyt*) because these EBV genomic sites have key roles in lytic reactivation[21] (Fig. 5c). To achieve rapid and robust lytic reactivation, we used a conditional P3HR-1 system, in which 4-hydroxy tamoxifen (4-HT) drives rapid nuclear translocation of the BZLF1 and BRLF1 proteins fused to a modified estrogen receptor that binds to 4-HT but not to estrogens in the media (P3HR-1 ZHT/RHT cells)[53]. We also added acyclovir to block synthesis of unchromatinized EBV genomes (Supplementary Fig. 6c, d), which could otherwise confound our analysis. Consistent with a potential latency maintenance role, H1.2 occupancy of the *BZLF1* promoter and leftward *oriLyt (oriLyt^L)* enhancer was reduced by ~30% and ~20%, respectively, at 6 h post-lytic reactivation (Fig. 5d, e). By 24 h post-lytic reactivation, H1 occupancy decreased by ~85% and ~90% at these sites (Fig. 5d, e). H1.2 occupancy was similarly reduced at the rightward *oriLyt (oriLyt^R)* enhancer, at the late gene *BcLF1* promoter, and at the promoter for latent membrane protein 1 (*LMP1*), which is expressed both in latency programs and as a lytic gene[54,55] (Fig. 5f and Supplementary Fig. 6e, f). To better determine if H1 association with chromatin was broadly reduced across the cell upon lytic reactivation, we also investigated the association of H1.2 with two human genome promoters: *GAPDH* (canonically euchromatic in Burkitt cells) and *MYO-D* (canonically heterochromatic in Burkitt cells). In latent cells, we observed lower levels of H1.2 association with the *GAPDH* promoter than with the *MYO-D* promoter, as expected (Supplementary Fig. 6g). Upon lytic reactivation, H1.2 association with both the *GAPDH* and *MYO-D* promoters decreased, and by 24 h post-lytic reactivation, this association fell to nearly undetectable levels (Supplementary Fig. 6g). These data suggest that the loss of H1 at the EBV genomic regions is most likely due to a global reduction in H1 chromatin occupancy upon EBV lytic reactivation, rather than a specific mechanism operative on the EBV genome.

We therefore next asked whether histone mRNA and protein abundances change with EBV lytic reactivation. Using RNA-seq (GEO GSE240008) and proteomic[56] datasets, we identified that *H1.2* and *H1.4* mRNAs were highly decreased by 12 h post EBV reactivation (Fig. 5g), whereas H1.2 protein levels decreased more rapidly than those of H1.4 (Fig. 5h). Notably, these differences did not appear to simply arise from host shutoff[57], as the mRNAs and protein levels of multiple octamer histones were not similarly decreased by EBV reactivation (Fig. 5g, h). These data raise the question of whether EBV evolved a mechanism to reduce H1 expression to reinforce the latency to lytic transition.

### EBV lytic reactivation alters SFPQ subnuclear distribution

EBV must ultimately circumvent host mechanisms to reactivate, including SFPQ and H1. Yet, our proteomic analysis did not identify decreased SFPQ protein abundance by 24 h post-reactivation[56]. To examine whether SFPQ protein levels might transiently be decreased at earlier stages of EBV lytic reactivation, we performed immunoblot time course analysis at four timepoints between 0 and 24 h. However, in contrast to H1.2 and H1.4, which were strongly decreased by 16 h post-reactivation (Fig. 6a), SFPQ protein levels remained stable (Fig. 6a). Since SFPQ can re-distribute into paraspeckle sub-nuclear bodies, which serve to sequester transcription factors and RNA-processing proteins[58–60], we hypothesized that EBV might alter SFPQ subcellular distribution and thereby also decrease H1 expression upon lytic reactivation.

To test this hypothesis, we followed SFPQ distribution by confocal microscopy in latency and during the first 24 h post-lytic reactivation. SFPQ exhibited a diffuse nuclear distribution with faint puncta prior to lytic induction. However, by 6 h post-P3HR-1 lytic reactivation by 4-HT addition, SFPQ redistributed into puncta, which reached near maximal levels by 16 h post-reactivation (Fig. 6b, c). The timing of SFPQ redistribution correlated with a reduction in H1 protein levels (Fig. 6a). Similar SFPQ redistribution was observed in three additional Burkitt cell lines by 24 h post-reactivation, all of which were lytically reactivated using distinct stimuli (Supplementary Fig. 7a, b). However, acyclovir blockade of EBV late gene expression did not preclude SFPQ redistribution into nuclear puncta (Supplementary Fig. 7c, d). Therefore, an EBV immediate early or early gene product may target SFPQ to nuclear puncta to reinforce the latency to lytic transition (Fig. 6d).

### SFPQ loss does not prevent initial EBV infection

We previously found that histone loaders are important both for the establishment and maintenance of EBV B-cell latency[16]. To investigate if SFPQ might also be important for EBV latency establishment, we expressed control or SFPQ sgRNAs in Cas9+ EBV-negative Ramos Burkitt cells and subsequently infected these cells with recombinant Akata strain EBV. Control and SFPQ depleted Ramos cells could both be infected by EBV at similar levels, as judged by expression of the Akata EBV GFP transgene (Supplementary Fig. 8a, b, d). Because both latent and lytic cycle proteins are expressed at the earliest stages of infection[61], we monitored via immunoblot whether SFPQ KO altered levels of the lytic proteins BZLF1 and BMRF1 or of the latency protein EBNA2, which is the first to be expressed upon nascent B-cell infection[62]. We observed decreased levels of both BZLF1 and BMRF1 at both 48 and 120 h post EBV infection of SFPQ KO cells as compared to control cells, even though H1.2 and H1.4 levels were substantially diminished by SFPQ loss (Supplementary Fig. 8c, e). In contrast, SFPQ KO increased EBNA2 levels at 48 h post infection, although this difference was diminished by 120 h post infection (Supplementary Fig. 8c, e). We likewise observed diminished leaky BZLF1 expression in SFPQ KO hTERT-immortalized normal oral keratinocytes (NOK) at 48 h post EBV infection, even though H1 levels were again strongly diminished by SFPQ loss (Supplementary Fig. 8f). Together, these data suggest SFPQ and histone H1 are not required for initial EBV infection or for the first steps of latency establishment, but that SFPQ and H1 may nonetheless impact the pattern of EBV gene expression at very early stages of infection.

## Discussion

EBV uses multiple latency programs to navigate the B-cell compartment and establish lifelong infection. Yet, much remains to be learned about the host factors and epigenetic mechanisms that enable EBV to

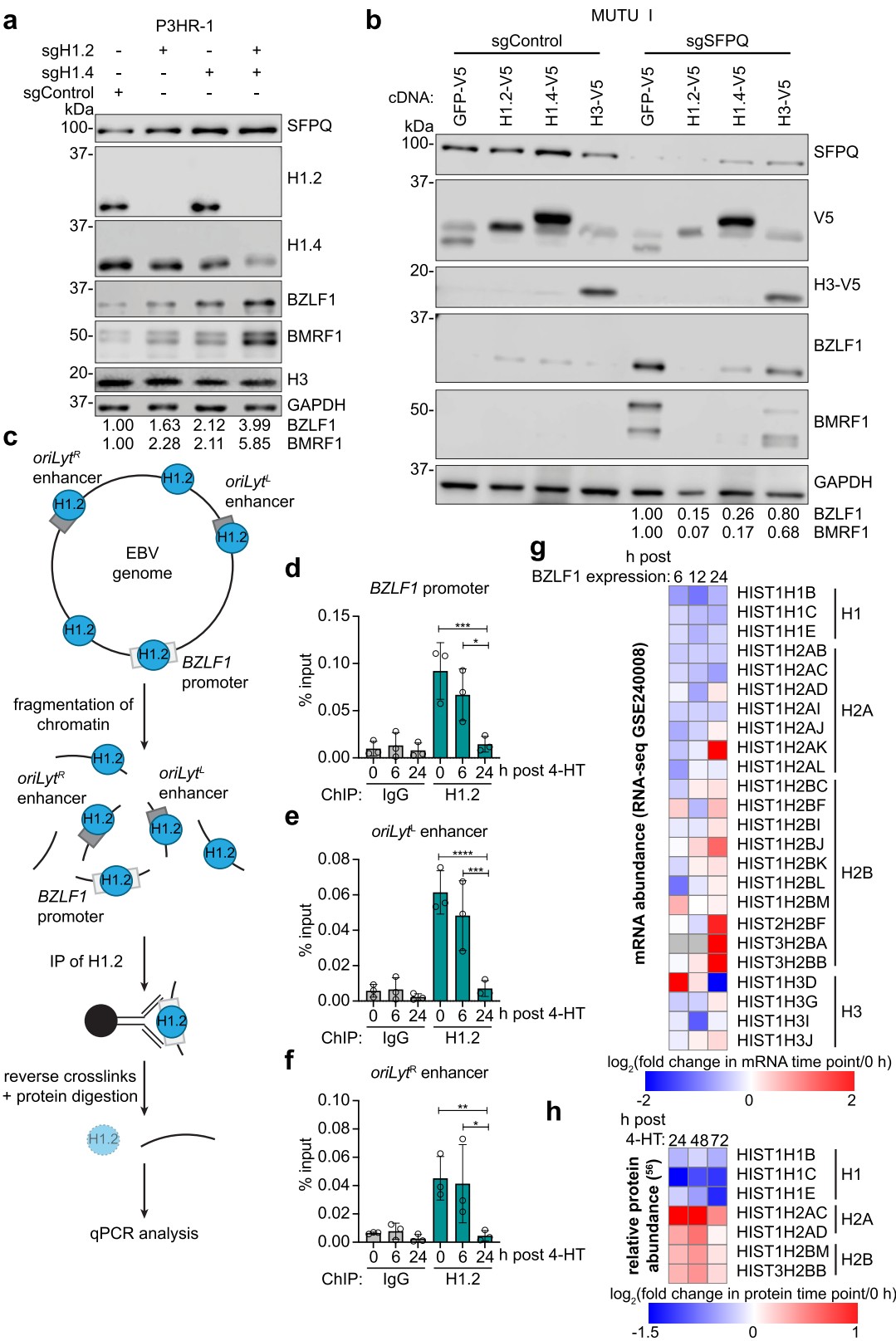

repress lytic gene expression. Here, we identified the nuclear protein SFPQ as critical for highly restricted forms of B and epithelial cell latency, including in the contexts of Burkitt lymphoma and gastric and nasopharyngeal carcinomas. Our results suggest that SFPQ is a driver of linker histone H1 expression in both EBV infected and uninfected cells, likely at the transcriptional level. In turn, H1 occupied multiple EBV genomic sites, including lytic reactivation control regions. Combined *H1.2* and *H1.4* knockout triggered lytic reactivation, whereas enforced H1 expression prevented reactivation even upon SFPQ loss. SFPQ was sequestered into nuclear puncta following reactivation, which correlated with reduced H1 abundance and EBV genomic occupancy.

**Fig. 5 | Histone H1 occupies multiple EBV genomic elements. a** Immuno-blot analysis of WCL from Cas9+ P3HR-1 cells expressing control, H1.2, and/or H1.4 sgRNAs. **b** Immunoblot analysis of WCL from Cas9+ MUTU I cells expressing V5-tagged GFP, H1.2, H1.4, or H3 cDNA and subsequently control or SFPQ sgRNAs. **c** Schematic of the H1.2 ChIP-qPCR assay. For clarity, nucleosomes are not shown. **d**–**f** ChIP-qPCR analysis of H1.2 occupancy at the EBV (**d**) *BZLF1* promoter, (**e**) *oriLyt*$^L$ enhancer, and (**f**) *oriLyt*$^R$ enhancer at 0, 6, and 24 h post 4-HT-induced lytic reactivation in Cas9+ P3HR-1 cells. These cells express the EBV immediate early proteins BZLF1 and BRLF1 fused to a modified estrogen receptor binding domain specific for 4-HT (P3HR-1 ZHT/RHT cells). 4-HT addition triggers BZLF1 and BRLF1 nuclear translocation and early gene expression. Mean ± standard deviation from $n = 3$ biological replicates is shown. *$p \leq 0.05$, **$p \leq 0.01$, ***$p \leq 0.001$, ****$p \leq 0.0001$ were calculated by one-way ANOVA. **g** Heatmap analysis of log$_2$(fold change) of histone mRNA abundance from RNA-seq analysis (GEO GSE240008) of MUTU I cells triggered for lytic reactivation by electroporation of a BZLF1 expression vector for the indicated times, relative to levels in cells electroporated for the same times with a GFP negative control expression vector. The values shown are for histones whose abundance was significantly decreased by SFPQ depletion in the RNA-seq analysis shown in Fig. 3. Histones are clustered by histone type. **h** Log$_2$(fold change) of histone protein abundance in P3HR-1 ZHT/RHT cells triggered for lytic reactivation by 4-HT for the indicated times, relative to levels in mock-induced cells[56]. The values shown are for histones whose transcripts were significantly decreased in abundance upon SFPQ depletion, relative to control levels, in the RNA-seq analysis shown in Fig. 3. Histones are clustered by histone type. Representative immuno-blots from $n = 3$ biological replicates and densitometry quantification with values normalized to the loading control GAPDH are shown. Source data are provided as a Source Data file.

Our studies implicate SFPQ as a key factor that supports H1 expression in both uninfected and infected B and epithelial cell contexts. While much remains to be learned about how linker histone H1 expression is controlled, SFPQ occupied the *HIST1H1C* promoter region in a DNA binding domain dependent manner. We therefore hypothesize that SFPQ plays a direct role in *HIST1* locus regulation. For instance, SFPQ associates with the RNA polymerase II (Pol II) C-terminal domain, and it may serve to recruit Pol II and/or a host cell transcriptional activator to the *HIST1* locus[63]. Alternatively, SFPQ has pleotropic roles in support of transcription-coupled splicing and mRNA 3'-end processing[63], and these may instead underlie its support of H1 expression. It is noteworthy that H1 variants are also major splicing regulators[64], and SFPQ may therefore crosstalk with H1 on multiple levels.

Our data are most consistent with a model in which a threshold level of total H1 is required to maintain EBV latency. Five of the six human histone H1 variants expressed in somatic cells are expressed from the *HIST1* locus in a cell replication dependent manner[19,65]. Of these, H1.2, H1.4, and H1.5 constitute >90% of the linker H1 expressed in lymphocytes[49], and the expression of each of these variants was significantly decreased by SFPQ KO. By contrast, CRISPR depletion of H1.2 or H1.4 alone was not sufficient to trigger EBV reactivation, likely because H1 isoforms share major functions[65], and residual H1 expression was sufficient to maintain latency. We speculate that combined H1.2 and H1.4 depletion likely crossed the H1 threshold required for EBV reactivation. This finding is reminiscent of the observation that B-cell lymphomas often have mutations within multiple H1 genes[65].

H1 is well positioned to epigenetically regulate EBV latency. Whereas core histones are stably incorporated into nucleosomes and require major energy input for their eviction[66,67], H1 binds less tightly and can therefore rapidly load and unload from DNA[68,69]. H1 residence times are an order of magnitude shorter than those of core histones[50,68,70]. Furthermore, H1 levels are dynamically regulated in cell differentiation[71]. Given that B and epithelial cell differentiation is a major trigger for EBV lytic reactivation, an intriguing possibility is that the EBV genome within latently infected cells senses changes in H1 abundance at the level of chromatin decompaction and gene de-repression[52,72,73]. In this manner, shifts in H1 abundance may serve as a readout of cell differentiation state that mechanistically alters viral genome accessibility and de-represses immediate early gene expression. While we favor a direct H1 role, our findings do not rule out the possibility that a threshold level of H1 is required to maintain expression levels of host genes important for maintaining latency.

Reversible chromatin epigenetic marks may add a further layer of regulation to EBV genomic H1 loading. For instance, H1 occupancy is enriched on promoters with repressive histone 3 lysine 9 trimethyl (H3K9me3) and histone 3 lysine 27 trimethyl (H3K27me3) marks[74–76], which are deposited on the *BZLF1* promoter and *oriLyt* enhancer in latency, but which are rapidly removed upon lytic reactivation[16,77,78]. Loss of H3K9me3 and H3K27me3 marks may serve as a signal for H1

unloading from these EBV genomic sites upon lytic reactivation. Once unloaded, H1 is targeted for proteasomal degradation[79], and this may have contributed to the rapid decline in H1.2 and H1.4 protein levels that we observed following lytic reactivation.

Akin to other herpesviruses[8,80], naked incoming double stranded DNA EBV genomes are organized into nucleosomes within the first 48 h post-infection[8]. However, despite their roles in maintenance of highly restricted forms of latency present in Burkitt lymphoma andnasopharyngeal and gastric carcinomas, our data suggest that neither SFPQ nor histone H1 play obligatory roles at the earliest stages of EBV latency establishment. Instead, we found that SFPQ KO reduced leaky lytic protein expression at 48 h post-infection in both Ramos B-cells and oral keratinocytes, even though H1 expression was reduced. This result may indicate that the roles of SFPQ and H1 in maintaining EBV latency become important after the establishment of upstream epigenetic mechanisms, including deposition of histone or DNA methylation marks or the formation of higher order EBV genomic architecture. However, depletion of SFPQ, and therefore also H1, may alter incoming EBV genomic epigenetic programing, potentially accounting for reduced leaky lytic protein and yet enhanced EBNA2 expression. A future goal will be to define how H1 loading dynamically changes over the phases of latency establishment, including as the EBV genome switches between latency programs.

SFPQ KO upregulated a large number of interferon stimulated genes, suggesting that SFPQ may directly repress their promoters[24–28,81]. Thus, SFPQ supports persistent EBV infection not only by maintaining viral latency, but also by suppressing antiviral responses. We suggest that both of these effects require H1. In support of this idea, RNAi knockdown of either the H1 chaperone TAF-1 or H1 itself was sufficient to upregulate ISGs in HeLa and 293 cells[82], putatively by altering ISG promoter structure. It will therefore be of interest to test whether enforced H1 expression blocks ISG induction in SFPQ KO cells. Future work will be needed to determine if depletion of H1 alters the expression of other host genes that in turn regulate the EBV lytic switch.

While we are not aware of other situations where H1 is required for double stranded DNA (dsDNA) viral latency, several dsDNA viruses target H1 within their lytic cycles. HSV-1 mobilizes H1 and core histones from chromatin within hours of African Green monkey Vero cell infection, a cellular environment that results in lytic replication rather than latency[83,84]. However, the significance of elevated free H1 pools in HSV-1 replication has remained unknown. The human papillomavirus type 11 E1 and simian virus 40 T-antigen each displace H1 from viral genomic origins to support lytic replication[85,86]. Similarly, the KSHV ORF59 viral DNA polymerase processivity factor binds to core and linker histones, which may play an important role in localizing the KSHV DNA polymerase to the viral genomic origin of lytic replication[87]. Furthermore, the adenovirus histone-like protein VII promotes H1 unloading to disrupt the host cell cycle[88]. While our data suggest that H1 is lost across both the host and EBV genomes upon lytic

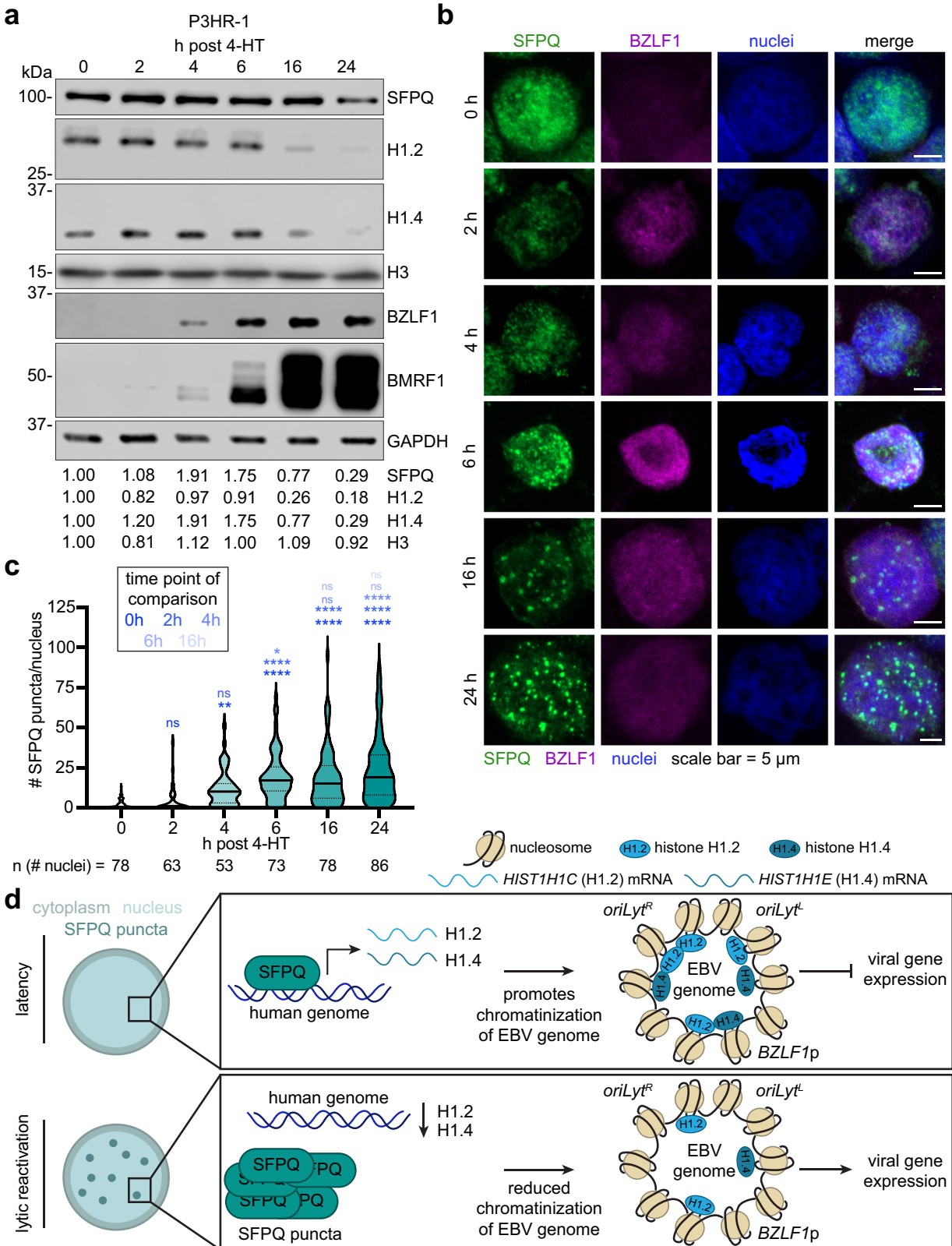

**b** SFPQ BZLF1 nuclei merge (time points: 0 h, 2 h, 4 h, 6 h, 16 h, 24 h)

SFPQ BZLF1 nuclei scale bar = 5 μm

reactivation, it will be interesting to determine whether EBV lytic cycle factors may also interact with H1 to support lytic replication.

How does EBV suppress histone H1 mRNA and protein expression following lytic reactivation? H1 loss is unlikely to have simply resulted from host shutoff, in which the EBV-encoded BGLF5 alkaline nuclease degrades a subset of host mRNAs[57,89], because most other histone mRNAs did not decline as robustly. Rather, our data instead suggest

that EBV-driven SPFQ sequestration underlies the observed H1 depletion. SFPQ sequestration was evident within hours after EBV immediate early gene activation but was not dependent on late genes, which suggests that an early gene may be responsible. However, we did not observe a protein-protein interaction between SFPQ and an EBV early gene product in our lytic cycle proteomic analysis[90]. While a false negative result is possible, we speculate that an EBV lncRNA, or an EBV-

**Fig. 6 | SFPQ re-distribution correlates with the temporal loss of H1.2 and H1.4 during EBV lytic reactivation. a** Immunoblot analysis of WCL from P3HR-1 cells at the indicated timepoints post EBV lytic reactivation by 4-HT addition. Representative immunoblot from $n = 3$ biological replicates and densitometry quantification with values normalized to the loading control GAPDH is shown. **b** Representative immunofluorescence images from $n = 3$ biological replicates of SFPQ (green), BZLF1 (magenta), or nuclear DAPI (blue) signals in P3HR-1 cells triggered for lytic reactivation by 4-HT for the indicated timepoints. Scale bar = 5 μm. **c** Quantification of the number of SFPQ puncta/nucleus during EBV lytic reactivation of cells treated as in (**b**). Violin plots show the median (solid line) and interquartile range (dotted lines). The number of nuclei quantified across three biological replicates are indicated. *$p \le 0.05$, **$p \le 0.01$, ****$p \le 0.0001$, ns not significant were calculated by one-way ANOVA. Values at a given timepoint were cross-compared with all subsequent timepoints. The color code indicated in the box at the top left indicates the timepoint used for cross-comparison for significance calculation. **d** Model of the roles of SFPQ and H1 in repressing EBV lytic reactivation. During EBV latency, SFPQ promotes expression of the histone H1 variants H1.2 and H1.4, which accumulate on the EBV genome at key regions and promote a compacted, repressive chromatin state that impedes expression of lytic genes. Upon EBV lytic reactivation, SFPQ is re-distributed within the nucleus to puncta by 6 h post lytic reactivation. This re-distribution precludes SFPQ from supporting H1.2 and H1.4 expression. Expression and protein levels of these H1 variants decline and so does their association with the EBV genome, which facilitates EBV lytic gene expression. Source data are provided as a Source Data file.

induced host lncRNA, may drive SFPQ redistribution. In support of this hypothesis, the host lncRNA *NEAT1* can redistribute SFPQ to phase-separated paraspeckles[91], although we were unable to obtain evidence that it re-localized SFPQ in lytic cells. Future work will be needed to determine whether the EBV *BHLF1* lncRNA, which is expressed with early kinetics and can be associated with sub-nuclear puncta[92], is instead required.

There is significant interest in triggering EBV lytic reactivation to sensitize EBV-associated tumors to destruction by the antiviral agent ganciclovir or to cytotoxic T-cells. Therefore, our data suggest that small molecules that block SFPQ/DNA interactions or protein degraders[93] that target SFPQ for proteasomal turnover could be intriguing therapeutic approaches across EBV-associated lymphomas and carcinomas, which collectively represent 1% of human cancer[2].

In summary, we identified that SFPQ and linker histone H1 were each necessary for EBV latency. SFPQ supports expression of H1, which occupied key EBV genomic regions during latency. Upon EBV lytic reactivation, SFPQ was sequestered in subnuclear puncta, which correlated with decreased global H1 expression and genomic occupancy.

## Methods
### Cell lines and culture
All B-cell lines were Cas9+. With the exception of the (EBV−) MUTU I and Ramos cell lines, all Cas9+ cell lines were previously generated. Briefly, the *Streptococcus pyogenes* Cas9 gene was introduced via lentiviral transduction followed by blasticidin selection, as has been reported previously[94]. The B-cell lines used in this study are as follows with the sources indicated: P3HR-1 ZHT/RHT Cas9+ (original cell line gift from Drs. Elliott Kieff and Eric Johannsen, Cas9+ cell line from ref. 95), MUTU I (EBV+) Cas9+ (original cell line gift from Dr. Jeff Sample, Cas9+ cell line generated in our lab), MUTU I (EBV−) Cas9+ (original cell line gift from Dr. Bill Sugden, Cas9+ cell line generated in this study), Akata EBV+ Cas9+ (original cell line gift from Dr. Elliott Kieff, Cas9+ cell line generated in ref. 21), Daudi EBV+ Cas9+ (original cell line from ATCC, CCL-213, Cas9+ cell line generated in ref. 95), Ramos Cas9+ (original cell line gift from Dr. Elliott Kieff, Cas9+ cell line generated in this study), and JSC-1 Cas9+ (gift from Dr. Kenneth Kaye). The other cell lines used in the study are from the following sources: C666.1 (gift from Dr. Elliott Kieff), SNU-719 (gift from Dr. Adam Bass), YCCEL1 (gift from Dr. Elliott Kieff), and hTERT-immortalized NOK (gift from Dr. Karl Munger). Versions with stable Cas9 expression were established in our lab. The B-cell lines and the nasopharyngeal carcinoma cell line C666.1 were cultured in Roswell Park Memorial Institute 1640 Medium (RPMI) (Life Technologies) supplemented with 10% fetal bovine serum (FBS) (v/v) (Thermo Fisher) and 1% penicillin-streptomycin (P/S) (v/v) (Life Technologies). Derivative cell lines were authenticated by immunoblotting, confocal microscopy, and/or T7E1 ligase assay, depending on the perturbation induced in the cell line. SNU-719 cells were cultured as above except that the P/S was omitted. YCCEL1 cells were cultured in Eagle's Minimum Essential Medium (EMEM) (ATCC) supplemented with 10% FBS (v/v). NOK cells were cultured in Keratinocyte-SFM 1X (Thermo Fisher). 293T cells were purchased from ATCC and cultured in Dulbecco's Modified Eagle Medium (DMEM) (Thermo Fisher Scientific) supplemented with 10% FBS and 1% P/S. All cell lines were grown at 37 °C and 5% $CO_2$ and were tested for mycoplasma using a MycoAlert Mycoplasma Detection Kit (Lonza). Cell line identity was authenticated via immunoblot for EBV positive or negative cell lines.

### Cloning and plasmid generation
To generate the plasmids for SFPQ, NONO, H1.2, H1.4, or H3 knock-out, sgRNA was cloned into pLenti-Guide-puro vector (Addgene #52963) or pLenti-Guide-zeo (made in our lab) by ligation after digestion with the BsmBI restriction enzyme. Control plasmids were either pXPR-011 (Addgene #59702) or pLenti-sg510-puro (generated in our lab). For knockout of BXLF1 or BZLF1, sgRNA was cloned into the pLenti-Guide-hyg vector (Addgene #62205) via a similar manner. To make the SFPQ cDNA rescue vector, SFPQ cDNA entry vector was purchased from DNASU (HsCD00516213). Site-directed mutagenesis using the NEB Q5 Site-Directed Mutagenesis Kit (E0554) and primers listed in Supplementary Data 2 were used to make a silent mutation at the PAM site targeted by SFPQ sg #1. The resulting DNA was subcloned into pLX-TRC313 (gift from Dr. John Doench) by digestion of the pLX-TRC313 vector with EcoRV and NheI, PCR amplification of SFPQ from the PAM-mutated pENTR223 vector using primers listed in Supplementary Data 2, and subsequent Gibson assembly. Additional SFPQ domain deletion mutants were generated by PCR amplification of the regions of interest and Gibson assembly (primers listed in Supplementary Data 2). Domain deletion mutants were cloned into the TRC313 vector by Gateway assembly or the pHAGE vector (gift from Dr. James DeCaprio) by PCR amplification of the SFPQ mutant from the respective TRC313 vector (primers listed in Supplementary Data 2) followed by Gateway recombination first into the pENTR223 vector and then into the pHAGE vector. For overexpression of H1.2, H1.4, and CTBP1, cDNA entry vectors were purchased from DNASU (HsCD00507003, HsCD00508384, and HsCD00511818) and cloned into pLX-TRC313 via Gateway recombination. For overexpression of histone 3, the histone 3 sequence was amplified from mTurquoise-H3-23 (Addgene #55558) and cloned into pLX-TRC313 via Gateway recombination (primers listed in Supplementary Data 2). The MYC overexpression vector that was used was previously generated in ref. 21. The pLX-TRC313-GFP and pHAGE-GFP vectors were previously generated in our lab.

### Stable cell line generation
All stable cell lines, unless otherwise stated, were generated via lentiviral transduction. Briefly, 293T cells were transfected with 150 ng VSV-G, 400 ng psPAX2, and 500 ng plasmid containing the gene to be overexpressed. Lentivirus was collected at 48 h and 72 h post-transfection and inoculated onto B-cells. Cells were selected for at least 6 days using the relevant selection marker (puromycin at 3 μg/ml, hygromycin at 5 ug/ml, or zeocin at 50 ug/ml). Overexpression was confirmed via immunoblot.

## CRISPR-based knockout

Transfection of 293T cells and transduction of B-cells, YCCEL1, SNU-719, C666.1, or NOK cells were performed as described above for stable cell line generation. Cells were selected for 3–6 days depending on the gene being knocked out and the selection maker (puromycin, hygromycin, or zeocin) being used. Knockout was confirmed by immunoblot or, as for BXLF1 KO, T7E1 ligase assay. Bulk populations of selected cells were used for downstream analyses.

## Induction of EBV lytic reactivation

Lytic reactivation was induced in various cell lines as follows: P3HR-1 ZHT/RHT (500 μM 4-HT), MUTU I (5 μg/ml IgM), Daudi (2 mM sodium butyrate plus 20 ng/ml TPA), and Akata (5 μg/ml IgG). Time points collected are indicated in the figures and legends.

## Immunoblot analysis

Samples were lysed in 1x Lammeli buffer by vortexing for 15 s and then heating at 95 °C for 10 min. Equal amounts of sample were loaded onto a tris-glycine SDS-polyacrylamide gel. Proteins were transferred to a nitrocellulose membrane, which was cut and blocked with 5% milk in 1x TBS. All primary antibodies were diluted into 1x TBST (0.1% Tween-20) and incubated overnight at 4 °C. The primary antibodies used are listed in Supplementary Data 3. The following day, membranes were washed with 1x TBST and incubated with secondary antibodies as listed in Supplementary Data 3 for 1 h at room temperature. Membranes were then washed with 1x TBST and, for those stained with HRP secondaries, incubated with SuperSignal™ West Pico PLUS Chemiluminescent Substrate (Thermo Fisher Scientific) for 2 min before imaging on a Li-Cor Odyssey Fc. Densitometry analysis was performed using ImageStudioLite Odyssey software (v 5.2.5) and samples were normalized to the loading control.

## Immunofluorescence microscopy

B-cells were collected, washed with 1x PBS, and dried on slides for at least 30 min at 37 °C. Cells were fixed with 4% paraformaldehyde for 10 min, washed twice with 1x PBS, permeabilized with 0.5% Triton X-100 for 5 min, and washed once with 1x PBS. Cells were then blocked for at least 1 h with 1% BSA. Cells were incubated in primary antibody for 1 h at 37 °C, washed with PBS, incubated with secondary antibody for 1 h at 37 °C, washed with PBS, incubated with 1:5000 Hoechst (Thermo Fisher) for 15 min at 37 °C, and then mounted using ProLong™ Diamond Antifade Mountant (Thermo Fisher). Samples were imaged on a Zeiss LSM 800 microscope using a 63x objective or 20x objective and collected as Z-stacks. Image analysis was performed in ImageJ. Number of SFPQ puncta were quantified using the 3D-object counter. Percentages of EBV or KSHV positive cells were counted by using the threshold tool and quantifying the number of objects stained for nuclei (total number of cells) or number of objects stained for either EBV or KSHV. Antibodies used are listed in Supplementary Data 3.

## RT-qPCR analysis

RNA was extracted using a RNeasy extraction kit with on-column DNA-digestion (Qiagen). RNA was reverse transcribed using iScript reverse transcription supermix (Bio-Rad). RT-qPCR was conducted using Power SYBR Green PCR mix (Applied Biosystems) on a Biorad CFX Connect Real Time System. Relative expression compared to an internal control (18S) was calculated using the deltadeltaCt method. Primers used for RT-qPCR are listed in Supplementary Data 2.

## Quantification of EBV genome copy number

To quantify intracellular EBV genome copy number, cells were harvested, and DNA was extracted using a Blood and Cell Culture DNA minikit (Qiagen). To quantify extracellular EBV genome copy number, supernatant was harvested. Supernatant was diluted at a 1:4 ratio into qPCR DNA Resuspension Buffer (400 mM NaCl, 10 mM Tris pH 8,

10 mM EDTA). Any un-encapsidated DNA was digested via incubation with DNAse I for 10 min at 37 °C. The enzyme was subsequently inactivated via heating at 75 °C for 10 min. Samples were incubated overnight at 37 °C with 40 μg/ml proteinase K and 0.16% SDS. Samples were phase extracted three times with 1:1 phenol:chloroform (v/v). The samples were then incubated overnight at −20 °C with 4 μl linear acrylamide, 1:10 (v/v) 3 M NaOAc, and 2 volumes ice cold 100% ethanol. Samples were centrifuged at $14,000 \times g$ for 10 min at 4 °C. The supernatant was decanted, and the pellet was washed with 1 ml 70% ice cold ethanol and centrifuged at $14,000 \times g$ for 2 min at 4 °C. The supernatant was decanted, and the pellet was dried overnight. The DNA pellet was resuspended in AE Buffer. Extracted DNA for both the intracellular and extracellular viral genomes was diluted to 10 ng/μl, and qPCR for the viral *BALF5* gene was performed on a Biorad CFX Connect Real Time System. A standard curve was made by serial dilution of the pHAGE-BLAF5 plasmid (previously generated in our lab) starting at 25 ng/μl. Copies of the viral genome were quantified using the linear regression equation generated from the standard curve. Primers used are listed in Supplementary Data 2.

## Flow cytometry analysis

For % gp350+ analysis, cells ($1 \times 10^6$) were harvested, washed twice with FACS buffer (2% FBS in PBS) and incubated at RT in the dark for 45 min with 1:1000 Cy5-gp350. Cells were washed twice with FACS buffer and then analyzed on a BD FACSCalibur. For % GFP+ analysis, cells ($1 \times 10^6$) were harvested, washed twice with FACS buffer, and analyzed on a BD FACSCalibur. Data analysis was performed using FlowJo X software.

## Cell cycle analysis

On day 2 post puromycin selection, cells were split to ensure that the cells were actively dividing at the time of collection. On day 3 post puromycin selection, cells were counted, and 1 million cells were fixed in 70% ethanol overnight at −20 °C. Samples were washed with 1x PBS, and then stained (0.2 mg/ml propidium iodide (PI, Thermo Fisher), 0.1% Triton, 0.1 mg/ml RNaseA) for 30 min at room temperature. Flow cytometry was performed on a BD FACSCalibur, and cell cycle analysis was performed using FlowJo X software.

## T7E1 ligase assay

The T7E1 ligase assay[96] was performed using the EnGenR Mutation Detection Kit (NEB #E3321) following the manufacturer's instructions. Briefly, DNA was extracted from B-cells using a Blood and Tissue Extraction Kit (Qiagen). The region around the expected CRISPR-mediated cut site in BXLF1 was amplified by PCR (primers listed in Supplementary Data 2) and purified via the QiaQuick PCR Purification Kit (Qiagen). The correct fragment size (~1000 bp) was confirmed via agarose gel electrophoresis. Equal amounts of control (sgControl) and BXLF1 KO (sgBXLF1) DNA (100 ng of each) were combined with 2 μl NEB2 buffer (total volume 19 μl) and heated to 95 °C for 10 min to denature the DNA strands and then cooled to 4 °C at 0.1 °C/s to allow for hybridization. Then, 1 μl of EnGen T7 Endonuclease I (New England Biolabs) was added, which will cleave DNA where distorted/mis-matched duplexes occur. This digestion reaction was incubated at 37 °C for 30 min. The products were run on a 2% agarose gel for visualization. The degree of gene modification was assessed by comparing the amount of cleaved DNA (lower two bands) to the amount of uncleaved DNA (upper band).

## RNA-seq library preparation and analysis

On Day 6 post expression of Control or SFPQ sgRNAs, 2 million unsynchronized Cas9 + P3HR-1 cells were collected in biological triplicate. RNA was extracted using a RNeasy extraction kit (Qiagen), and an on-column DNA digestion step was performed to remove possible contamination from genomic DNA. RNA quality (RIN score >7) was confirmed via Bioanalyzer analysis. For library preparation, 1 μg of RNA

was used for each sample, rRNA-depletion using the NEBNext rRNA Depletion Kit v2 (#E7400, New England Biolabs) was performed, and library construction was completed using NEBNext Ultra RNA Library Prep Kit for Illumina (#E7760L, New England Biolabs) as per the manufacturer's instructions. Library quality was confirmed via Bioanalyzer analysis. Libraries were multi-indexed, pooled, and sequenced on an Illumina NextSeq 500 sequencer using single-end 75-bp reads (Illumina) at the Molecular Biology Genomics Core Facility at the Dana-Farber Cancer Institute. Single-end reads were mapped to the hg19 human (GRCh37) and Akata EBV genomes. Salmon (v1.0.0) was used to quantify the transcripts[97], and DESeq (v1.14.1)[98] was used to determine differentially expressed genes. Genes that had a log$_2$(fold change) greater than 1 (actual fold change greater than 2) and an adjusted *p* value of <0.05 were considered significant. Heatmaps were generated using Morpheus, and Enrichr was used to conduct gene ontology enrichment analysis.

## ChIP-qPCR analysis

For quantification of H1.2 accumulation on the EBV genome, P3HR-1 cells were treated with 500 μM 4-HT to induce lytic reactivation and 100 μg/ml acyclovir (Millipore) to prevent accumulation of replicated, linear genomes and collected at the indicated time points. Cells were harvested and cross-linked with 1% formaldehyde for 10 min at room temperature with rotation before the reaction was quenched with a final concentration of 0.3 M glycine for an additional 5 min. Cells were washed twice with cold 1x PBS and stored at −80 °C. Cells were lysed in lysis buffer (50 mM Tris pH 8, 10 mM EDTA, 1% SDS, protease inhibitor (Sigma)) for 20 min on ice. P3HR-1 cell lysate was sonicated for 10 cycles (30 s on/30 s off) in a Diagenode water bath-sonicator. After confirmation of correct fragment size via agarose gel electrophoresis, samples were centrifuged at 14,000 × *g* for 10 min at 4 °C. Soluble chromatin was diluted 1:10 in ChIP dilution buffer (16.7 mM Tris pH 8, 1.2 mM EDTA, 167 mM NaCl, 1.1% Triton X-100, 0.01% SDS, and protease inhibitor). The equivalent of chromatin from 1 million cells was combined with 2 μg anti-H1.2 (Proteintech 19649-1-AP) or anti-IgG (Cell Signaling 2729S) antibodies and 20 μl pre-washed magnetic protein A/G beads (Thermo) and incubated overnight with rotation at 4 °C. The same protocol was followed for the HA-SFPQ ChIPs except for the following modifications: the MUTU I cells were sonicated for a total of 20 cycles, and the equivalent of chromatin from 10 million cells was combined with 10 μl anti-HA antibody (Cell Signaling C29F4) or an equivalent μg amount of anti-IgG antibody and 50 μl pre-washed magnetic protein A/G beads. After overnight incubation, beads were washed for 5 min of rotation each using the following series: twice with low salt buffer (20 mM Tris pH 8, 2 mM EDTA, 150 mM NaCl, 1% Triton X-100, 0.1% SDS), twice with high salt buffer (20 mM Tris pH 8, 2 mM EDTA, 500 mM NaCl, 1% Triton X-100, 0.1% SDS), once with LiCl buffer (10 mM Tris pH 8, 1 mM EDTA, 0.25 M LiCl, 1% NP40, 1% sodium deoxycholate), and once with TE (IDT). For elution and to reverse the cross-links, beads were resuspended in 100 (H1.2 ChIP) or 200 (HA ChIP) μl elution buffer (100 mM NaHCO$_3$, 1% SDS), supplemented with 5% v/v Proteinase K (NEB), and incubated at 65 °C for 2 h with shaking at 400 rpm. Eluted DNA was purified using the QIAquick PCR purification kit (Qiagen). Samples were quantified using qPCR run on a Biorad CFX Connect Real Time System and normalized to the percentage of input DNA (% input). Primers used are listed in Supplementary Data 2.

## Akata virus production

Akata cells were induced with 0.2% (v/v) rabbit IgG (Dako) for 6 h at 37 °C. The media was then changed to fresh RPMI supplemented with 4% FBS, and cells were cultured for 3 days. Viral supernatant was then collected via filtration through a 0.45 μm filter and concentrated 100-fold via centrifugation at 50,000 × *g*. Virus was stored at −80 °C until used.

## Ramos and NOK cell infection

For Ramos cells, SFPQ KO was performed as described above. On day 1 post puromycin selection, 1 million cells were infected with Akata GFP+ virus in a total volume of 200 μl. Two hours after initial infection, the total volume was raised to 1 ml with RPMI media supplemented with 10% FBS and 1% P/S. Samples were collected for flow cytometry and immunoblot on day 3 post puromycin selection (48 h post infection) and day 6 post puromycin selection (120 h post infection). For NOK cells, SFPQ KO was performed as described above. Two days post puromycin selection, 0.1 million NOK cells were infected with 800 μl Akata EBV virus. Cells were incubated with the virus for 12 h before being washed once with RPMI and replacing the media with Keratinocyte-SFM. Samples were collected for immunoblot 48 h post infection.

## Statistical analysis

GraphPad Prism v. 10.0.03 was used for all statistical analyses unless otherwise stated. Bar graphs depict the mean plus or minus standard deviation unless otherwise stated. The number and type of replicates are included in the figure legends. Statistical significance tests were conducted using a one-way ANOVA with a Tukey multiple comparisons test unless otherwise stated. Statistical significance is indicated by asterisks in the figures: *$p \leq 0.05$, **$p \leq 0.01$, ***$p \leq 0.001$, ****$p \leq 0.0001$.

## Reporting summary

Further information on research design is available in the Nature Portfolio Reporting Summary linked to this article.

## Data availability

The RNA-seq data generated in this study (SFPQ KO) have been deposited in the NIH GEO database under accession code GSE235265. The RNA-seq data (lytic time course) used in this study are available in the NIH GEO database under accession code GSE240008 and the proteomic data used in this study are available in the ProteomeXchange Consortium PRIDE database under accession code PXD006317. The immunoblot and *p* value data generated in this study are provided in the Source Data file. All figures were made using commercially available GraphPad, Adobe Illustrator, or BioRender. Specifically, Figs. 1c, 3a, and 4a were created entirely or in part with BioRender.com released under a CC-BY-NC-ND license. Source data are provided with this paper.

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

## Acknowledgements

This work was supported by NIH T32 AI007245 and NIH F32 AI172329 to L.A.M.N., by NIH T32 AI007245 and an American Cancer Society Post-doctoral Fellowship to E.M.B., by a Lymphoma Research Foundation Postdoctoral Fellowship to Y.L., by a Special Fellow award from the Leukemia & Lymphoma Society to N.A.U., by NIH K99/R00DE031016 to R.G., and by NIH R01 CA275301, P01 CA269043, R01 AI164709, R01 CA228700, and R21 AI170751 and a Burroughs Wellcome Career Award in Medical Sciences to B.E.G. RNA-seq datasets (GEO GSE240008) were generated with support from R01 CA272142. We thank Dr. Erik Flemington for helpful discussions. We thank members of the Gewurz and Zhao labs for helpful discussions. We are grateful to Dr. Eric Johannsen for P3HR-1 ZHT/RHT cells, Dr. Bill Sugden for the MUTU I (EBV-) cells, Dr. Karl Munger for the Ramos cells, and Dr. Jaap Middeldorp for the anti-BMRF1 antibody (OT14E2). We are thankful for the support of the Molecular Biology Genomics Core at the Dana Farber Cancer Center for RNA-seq data acquisition.

## Author contributions

L.A.M.N and B.E.G. designed experiments. L.A.M.N., C.L., E.M.B., Y.L. and N.A.U. performed experiments. L.A.M.N., C.L., E.M.B., N.A.U., R.G. and B.E.G. analyzed data. L.A.M.N. and B.E.G. wrote the manuscript.

## Competing interests

The authors declare no competing interests.
