## [Peer Review File · Nature Communications]

The nucleic acid binding protein SFPQ represses EBV lytic reactivation by promoting histone H1 expressionREVIEWER COMMENTS

Reviewer #1 (Remarks to the Author):

Murray-Nerger et al present an interesting and well written study focused on how host protein SFPQ is critical for EBV reactivation. Using an elegant series of experiments and clear hypotheses, the authors show that KO of SFPQ leads to a decrease in H1 expression and subsequent reactivation of EBV. While the experiments presented are of high significance as the role of heterochromatin formation on EBV genomes is of great interest to the field, there are some major concerns with respect the model proposed. Most importantly, H1 binding to DNA without nucleosomes present, as is proposed for the EBV genome, is difficult to imagine. Only in exogenous systems (eg xenopus extracts) has H1 been reported to bind DNA without nucleosomes, and much more evidence would need to be provided to suggest that no core histones/nucleosomes are important to H1-supported EBV genome compaction. Specific comments are below:

- BXL1 KO – this is not clear to a non-EBV audience. It would be helpful to add labels like IE, E or L to the figure to make it easier to follow for a broad readership.
- It would be helpful to know about SFPQ homodimers in context in the introduction. How does this support the idea that SFPQ affects histone mRNA stability? These mechanistic insights would be helpful to the reader to understand what might be happening.
- How does cell cycle affect the action of SFPQ? G2/M transition was also an enriched category in the GO analysis. If the cells are transitioning from one stage to another as a reaction to SFPQ KO, then the conclusion that SFPQ specifically affects H1 is missing the possibility that it is an indirect effect of cell cycle misregulation. Perhaps EBV still takes advantage of this, but so far it is not conclusive. Is SFPQ bound to the EBV genome? ChIP of SFPQ?
- Line 203-205: The speculation that decreased expression of histone genes does not necessarily mean that DNA is exposed. DNA is exposed during S phase but that does not trigger ISG expression. Again, this could be cell cycle dependent. Were the cells in the SFPQ KO RNA-seq experiment synchronized at any point?
- The fact that SFPQ KO decreases H1 expression in the absence of EBV is an important finding relevant to multiple fields. However, it is unclear whether that is a direct or indirect effect, and whether any effect on EBV is an attempt of the cell to silence the invading genome or a mechanism exploited by the virus to ensure persistence in latency. This should be discussed.
- Fig 4 conclusions – it is a nicely conducted experiment to determine that the DBD of SFPQ is required for maintaining latency. However, it could be binding the H1 gene locus through this domain, and regulating H1 transcription which in turn affects EBV, though not directly.
- sgRNAs generally cause deletion, but seeing as there is not full KO with many of the experiments presented (like H1.4), it is likely that within the population, some cells lose H1.4 and others do not. How do the authors consider this? Population effects should be discussed.
- In Figure 5, the overexpression of H3 is used as a control of H1 isoform overexpression to show that H3 does not have an effect. However, this is perhaps not the best control for H1 since H3 requires H4 to form a tetramer, plus chaperones to be deposited on DNA. It might be more informative to overexpress HP1 or another heterochromatin marker to determine whether the H1-associated suppression of reactivation is specific to H1/SFPQ or any kind of heterochromatin.
- Can the authors rule out the possibility that the cellular reaction to depletion of H1 is a stress response of the cell that then results in modification/expression of a different gene set that then triggers reactivation? While H1 is ChIP-ed to the EBV genome under the conditions presented, that does not negate the possibility that host factors are also affected and may promote reactivation otherwise.
- It is not clear how the authors resolve a model where H1 is loaded onto EBV genomes but core histones are not. As described, core histones make up the nucleosome and H1 linker histones bind to the DNA at the entry/exit sites of the nucleosome. That means H1 binds the nucleosome, and in fact, was described to be unable to bind the nucleosome without linker DNA. The affinity of H1 for DNA without histones on it is not discussed here, but suggested in the model where H1 promotes latency, and therefore compaction of the EBV genomes, without the presence of core histones/nucleosomes.

Reviewer #2 (Remarks to the Author):

In this manuscript, Murray-Nerger et al. identify a novel role for splicing factor proline and glutamine rich (SFPQ) in Epstein-Barr virus (EBV) latency. The authors previously performed a genome-wide CRISPR/Cas9 loss-of-function screen in Burkitt lymphoma EBV+ B-cells to identify factors that regulate EBV latency (Mol Cell (2020) 78(4):653-669.e8. <https://doi.org/10.1016/j.molcel.2020.03.025>). In this submitted manuscript, the Authors characterized a role for SFPQ, a top hit from the CRISPR/Cas9 screen, in EBV latency. Using independent sgRNAs targeting SFPQ, the authors identify a role for SFPQ in EBV latency maintenance in multiple B- cell and epithelial cell lines. Moreover, the Authors show that SFPQ regulates latency across EBV types (I and II) and for another gammaherpesvirus, KSHV. SFPQ represses EBV lytic replication at the level of IE gene expression through a mechanism that is distinct from that of Myc, another top hit from the CRISPR/Cas9 screen. RNAseq analysis highlighted a role for SFPQ in the regulation of histone gene expression, particularly for linker histones H1.2 and H1.4. In absence of SFPQ, linker histones levels were depleted, H1 occupancy on the EBV genome decreased, and EBV lytic replication initiated. sgRNA depletion of H1.2 and H1.4 induced EBV lytic replication, whereas enforced H1.2 and H1.4 expression prevented EBV lytic reactivation in absence of SFPQ. The Authors identify that SFQP regulation of EBV latency requires its DNA-binding-domain, but neither RNA recognition motif. The levels of SFPQ remain stable following EBV lytic reactivation, although SFPQ progressively redistributes into nuclear puncta, which correlates with reduced H1 expression. This manuscript is clearly written, the data is clearly presented, convincing, and largely supports the Authors' conclusions. The submitted work highlights several novel findings, including previously unidentified roles for SFPQ and H1 in EBV latency.

Specific comments:

The relationships between SFPQ, H1, and EBV latency maintenance are convincingly characterized and nicely presented. However, it is not clear whether SFPQ directly or indirectly regulates H1 expression or association with EBV genomes. The Authors present and suggest a model in which SFPQ regulates H1 expression, likely at the H1 transcriptional level due to the requirement for the SFPQ DBD. However, given that H1.2 and H1.4 are DNA-replication-dependent histones within the HIST1 histone cluster, it is not clear how SFPQ regulation of H1 expression to maintain EBV latency conforms with the cell-cycle dependent expression of these histones. Experiments to show that SFPQ associates with H1.2 or H1.4 promoters would strengthen the Author's model that SFPQ supports H1 transcription.

The SFPQ DBD is necessary for SFPQ to maintain EBV latency. The Authors propose that this is likely due to regulation or support of H1 transcription. Is it possible that SFPQ directly associates with latent EBV genomes to prevent lytic reactivation? Evaluation of whether or not SFPQ associates with latent EBV genomes would further inform the Authors model of SFPQ-mediated maintenance of EBV latency.

Depletion of SFPQ results in decreased H1.2 and H1.4 levels, which is associated with the loss of H1.2 and H1.4 at EBV genomic elements. Is the loss of H1 at these elements an active process or consequent to a global reduction in H1 levels. Is H1 similarly depleted on cellular promoters? Is it possible that SFPQ may facilitate or specifically promote stable association of H1 with EBV latent chromatin?

Minor comments:

In some instances, terminology used is not correct. For example, lines 58/59 "Early genes drive lytic replication of EBV genomes...". Lines 296/297 "an EBV immediate early or early gene may therefore target SFPQ...". "Gene" should be "protein". The Authors should ensure that the terminology used is appropriate.

Supplementary Figure 1 caption does not match the data presented in the Figure. The text describing Supplementary Figure 3 describes IFIT1 and IRF7 expression in P3HR1 and MUTU I cells (lines 200-201), yet the Figure presents IRF7 expression only in P3HR1 cells. The Authors should

ensure that text and accompanying figure captions appropriately describe what is presented in the Figure.

In some instances, information for figures is differently presented. For example, some immunoblots have densitometry quantification (such as Figure 4 b, d; Figure 5 b) whereas others do not (such as Figure 4 g; Figure 5 a). The Authors should ensure that data is presented consistently.

Reviewer #3 (Remarks to the Author):

This manuscript reports a role for the host SFPQ protein in maintaining EBV latency. The Gewurz lab previously identified SFPQ as a strong hit for genes that help suppress EBV replication in the EBV+ P3HR1 Burkitt lymphoma cell line. Here, they report follow-up experiments, demonstrating its role in multiple EBV+ lymphoma and carcinoma cell lines, and dissecting the mechanism by which SFPQ suppresses EBV replication. They provide evidence that SFPQ acts independently (or possibly downstream) of c-myc - which they had previously reported contributed to maintenance of latency. To further define the mechanism, they perform a CRISPR knockout of SFPQ in P3HR1 cells and find levels of many interferon regulated mRNAs are increased and Histone H1 mRNAs are decreased. They further demonstrate that these Histone H1 proteins bind to regulatory loci on the EBV genome and, most significantly, H1.2 and H1.4 redistribution is temporally associated with lytic reactivation in P3HR1 cells.

The experiments in the manuscript are of high quality and well controlled. Understanding the mechanisms by which EBV remains in a latent state is of fundamental importance, but also has significant implications for translational efforts to treat EBV+ cancers with "lytic induction therapy." Understanding these mechanisms may improve treatment efficacy and help overcome resistance. Additional experiments are not required to convince me that SFPQ plays a role, via Histone H1 in maintaining EBV latency in various EBV+ tumors. However, there is nothing in this manuscript that convinces me that SFPQ is important for establishment of latency upon initial EBV infection (implied in several places, most notably lines 311-327). The experiments presented here rely almost exclusively on a single BL cell line, P3HR1, although Fig 1D does establish the importance of SFPQ in multiple other EBV+ BLs and carcinomas. What is not established is whether SFPQ is important for EBV latency in normal (non-cancer) cells. EBV+ cancers are under strong immune selection pressure against expression of EBV lytic genes. As the authors point out, B cell lymphomas frequently have mutations in multiple histone H1 genes (lines 337-338). It should not be assumed that SFPQ's ability to regulate histone H1 genes to enforce EBV latency is the normal situation and did not arise from similar selection pressure. This remains an interesting unanswered question.

We sincerely thank the reviewers for their thorough review of our manuscript and for their comments and recommendations. Based on the points raised by the reviewers, we have performed additional experiments coupled with text and figure changes to address the reviewers' concerns. Specifically, we expanded our analysis of SFPQ DNA-binding domain roles in target host and viral gene regulation and now show that the DBD supports SFPQ association with host genes, including the H1.2 promoter, and with multiple EBV genomic regions. We clarify our model that linker H1 binds the EBV genome in the context of core nucleosomes. We confirm key SFPQ roles in histone H1 regulation even in EBV uninfected cells, suggesting that this is not an adaptation by EBV to escape immune pressure but rather a general mechanism that supports histone H1 expression, which is in turn necessary for maintenance of the EBV latency I program. We expanded our study to interrogate the role of SFPQ upon nascent EBV infection and establishment of latency. As a result of these revisions, we have added 2 new figure panels in the main manuscript (Figures 4h, 6d) and 21 new supplementary figure panels (Supplementary Figures 4a-f, 5c-d, 5g-k, 6b, 6g, 8a-f). Importantly, these additional data support our original conclusions, and we feel that the manuscript has been strengthened by the revision. Here we present a point-by-point answer to the reviewers' comments. Our responses are marked with ">Response:".

REVIEWER COMMENTS

Reviewer #1 (Remarks to the Author):

Murray-Nerger et al present an interesting and well written study focused on how host protein SFPQ is critical for EBV reactivation. Using an elegant series of experiments and clear hypotheses, the authors show that KO of SFPQ leads to a decrease in H1 expression and subsequent reactivation of EBV. While the experiments presented are of high significance as the role of heterochromatin formation on EBV genomes is of great interest to the field, there are some major concerns with respect the model proposed. Most importantly, H1 binding to DNA without nucleosomes present, as is proposed for the EBV genome, is difficult to imagine. Only in exogenous systems (eg xenopus extracts) has H1 been reported to bind DNA without nucleosomes, and much more evidence would need to be provided to suggests that no core histones/nucleosomes are important to H1-supported EBV genome compaction. Specific comments are below:

- BXL1 KO – this is not clear to a non-EBV audience. It would be helpful to add labels like IE, E or L to the figure to make it easier to follow for a broad readership.

>Response: We appreciate the reviewer helping to make our manuscript more accessible to a non-EBV audience. We have followed the reviewer's suggestion and clarified in both the figure (Fig. 2d) and figure legend what BXL1 and BZLF1 are.

- It would be helpful to know about SFPQ homodimers in context in the introduction. How does this support the idea that SFPQ affects histone mRNA stability? These mechanistic insights would be helpful to the reader to understand what might be happening.

>Response: We thank the reviewer for suggesting the addition of information about SFPQ homodimerization when SFPQ is first introduced. As with all DBHS family proteins, SFPQ is believed to form an obligate dimer, and this dimeric state is necessary for SFPQ function. SFPQ uses its coiled-coil domain to form either a heterodimer with another DBHS protein or a homodimer. SFPQ's ability to dimerize is required for its nucleic acid binding capabilities. We have included additional text in the introduction (p. 4) and added one new reference (Lee, et. al, *NAR* 2015) to better provide background about this important aspect of SFPQ biology.

- How does cell cycle affect the action of SFPQ? G2/M transition was also an enriched category in the GO analysis. If the cells are transitioning from one stage to another as a reaction to SFPQ KO, then the conclusion that SFPQ specifically affects H1 is missing the possibility that it is an indirect effect of cell cycle misregulation. Perhaps EBV still takes advantage of this, but so far it is not conclusive. Is SFPQ bound to the EBV genome? CHIP of SFPQ?

>Response: We appreciate the reviewer closely examining the data and pointing out the enrichment of the G2/M transition as a GO category that was significantly decreased upon SFPQ KO. The G2/M transition category was only enriched in the comparison of SFPQ sgRNA #1 to the control, while terms related to chromatin and nucleosome assembly were enriched in the comparisons for both guides. We initially focused only on terms that were enriched in experiments with both SFPQ guides. However, to address the reviewer's question, we performed cell cycle analysis in MUTU I EBV+ Cas9+ cells transduced with control or independent SFPQ guide RNAs. We also performed cell cycle analysis of MUTU I EBV+ Cas9+ cells that had been lytically reactivated via IgM stimulation for 24 hours. As expected, control cells had a high percentage of S phase cells – Burkitt cells are amongst the most rapidly proliferating tumor cells (Pajic et al., *Int. J. Cancer* 2000 PMID: 10956386). However, we observed no significant difference in the percentage of G1 or G2 cell populations in control vs. SFPQ KO cells, suggesting that SFPQ KO does not affect the G2/M transition. While we did observe a modest decrease in the percentage of SFPQ KO cells in S phase compared to the sgControl at the early timepoint examined (same time point that was used for most of our experiments), we observed a more dramatic decrease in the percentage of S phase cells in the lytically reactivated cells. This observation suggests that the modest change that we see in the percentage of cells in S phase in the SFPQ KO cells is likely due to the induction of lytic reactivation that occurs upon SFPQ KO, rather than the loss of SFPQ itself, at this timepoint.

This finding is consistent with the published literature that has shown that the production of the immediate early EBV protein BZLF1 causes cell cycle arrest in G1 (Cayrol and Flemington, *The EMBO Journal* 1996 PMID: 8654372; and Rodriguez et al., *J. Virol.* 1999 PMID: 10516009). We have now incorporated these new data into Supplementary Fig. 4 and within the text (p. 10). Furthermore, the observation that enforced H1 expression rescues EBV latency (inhibits lytic reactivation), even when SFPQ is knocked out, supports our hypothesis that SFPQ supports EBV latency by supporting H1 expression, rather than by effects on cell cycle alone.

>To address the reviewer's second question about whether SFPQ binds the EBV genome, we performed ChIP-qPCR analysis, employing both HA-tagged full length (FL) vs. DNA binding

domain deletion mutant (Δ DBD) SFPQ constructs. SFPQ occupied multiple EBV genomic sites, including the *BZLF1* promoter, *oriLyt* and the C promoter (Cp) in a DNA-binding domain dependent manner. As the DNA binding motif of SFPQ is poorly characterized, these findings are intriguing for future work to better understand the interaction between SFPQ and target gene sites. We incorporated these data into Supplementary Fig. 5h-k and in the text on p. 11-12.

- Line 203-205: The speculation that decreased expression of histone genes does not necessarily mean that DNA is exposed. DNA is exposed during S phase but that does not trigger ISG expression. Again, this could be cell cycle dependent. Were the cells in the SFPQ KO RNA-seq experiment synchronized at any point?

>Response: We agree with this point and have modified the text to address this concern (p. 10). We interrogated our RNAseq data to investigate whether SFPQ KO might de-repress endogenous retrovirus expression as an alternative means for triggering ISG expression but did not find evidence of that as a potential mechanism. We now clarify in the methods section on p. 27 that we did not synchronize cell cycle.

- The fact that SFPQ KO decreases H1 expression in the absence of EBV is an important finding relevant to multiple fields. However, it is unclear whether that is a direct or indirect effect, and whether any effect on EBV is an attempt of the cell to silence the invading genome or a mechanism exploited by the virus to ensure persistence in latency. This should be discussed.

>Response: We appreciate the reviewer's interest in this important point. We now present RT-qPCR and immunoblot data to show that SFPQ knockout reduces H1 expression even in three EBV-negative cell models: Ramos B cells, normal oral keratinocytes and MUTU I Burkitt cells that have been cured of the EBV episome (Supplementary Fig. 5a, b, c, d). We also present ChIP-qPCR data to demonstrate that SFPQ occupies the H1.2 promoter (Fig. 4h, Supplementary Fig. 5g and revised text on p. 11-12). Given that H1 knockout is sufficient to induce EBV lytic reactivation, and since enforced H1 expression maintains EBV latency upon SFPQ KO, we hypothesize that SFPQ supports H1 expression, and this in turn is required for maintenance of the EBV latency I state.

The reviewer raises an interesting point about the possibility that the association of H1 with the EBV genome could be promoted by the host or the virus. Since EBNA1 is the only EBV encoded protein expressed in latency I, we speculate that EBV evolved to coopt histone H1 biology as a means to maintain this highly restricted latency state. We added to our discussion (p. 18-19) about these topics.

- Fig 4 conclusions – it is a nicely conducted experiment to determine that the DBD of SFPQ is required for maintaining latency. However, it could be binding the H1 gene locus through this domain, and regulating H1 transcription which in turn affects EBV, though not directly.

>Response: We completely agree with the reviewer that it is most likely that SFPQ is binding the H1 gene locus through the DNA-binding domain (DBD) and thus regulating H1

transcription, which subsequently affects the H1 levels and therefore EBV genome occupancy. As discussed above, SFPQ KO decreases H1.2 and H1.4 mRNA and protein levels (Fig. 4b-e). We now include ChIP-qPCR analysis with HA-tagged full length (FL) and the DNA binding domain deletion mutant (Δ DBD) SFPQ constructs. These data demonstrate that SFPQ occupies the H1.2 promoter in a SFPQ DBD dependent manner. We added this data as new panels in Fig. 4h and Supplementary Fig. 5g and revised the text on p. 11-12.

- sgRNAs generally cause deletion, but seeing as there is not full KO with many of the experiments presented (like H1.4), it is likely that within the population, some cells lose H1.4 and others do not. How do the authors consider this? Population effects should be discussed.

>Response: We agree that our CRISPR editing caused functional bi-allelic deletion of genes including H1.4 in many but not all cells. This is virtually always the case with CRISPR editing, since the NHEJ pathway occasionally restores Cas9 mediated double strand breaks at some frequency. We would therefore have had to obtain single cells to have a full KO across the population, but this is not feasible since EBV reactivation blocks proliferation. As immunoblot captures the bulk population of cells, this likely accounts for the substantial but not total loss of H1.4. Furthermore, in the case of H1.4, the weaker selection reagent (zeocin instead of puromycin) was used so that a double knockout could be accomplished (with puromycin selection being used for H1.2). In this case, it is possible that some cells were not transduced and escaped antibiotic selection. We clarified in our methods (p. 23) that we used bulk populations of cells.

- In Figure 5, the overexpression of H3 is used as a control of H1 isoform overexpression to show that H3 does not have an effect. However, this is perhaps not the best control for H1 since H3 requires H4 to form a tetramer, plus chaperones to be deposited on DNA. It might be more informative to overexpress HP1 or another heterochromatin marker to determine whether the H1-associated suppression of reactivation is specific to H1/SFPQ or any kind of heterochromatin.

>Response: We appreciate the reviewer's concern over using H3 as a control. We wish to emphasize that given the high level of histone abundance, it is unlikely that the level of H3 overexpression that we achieved would lead to an imbalance in the H3/H4 ratio in the nucleus. However, to address this concern, we repeated this experiment using overexpression of the repressor CTBP1, which we and others have found to be active in Burkitt cells. We generated a stable MUTU I EBV+ cell line expressing V5-tagged CTBP1 and subsequently knocked out SFPQ. In a manner similar to H3 overexpression, overexpression of CTBP1 failed to rescue the phenotype and inhibit lytic reactivation, in contrast to H1.2 or H1.4 overexpression. We included this expanded data in Supplementary Fig. 6b and p. 12-13.

- Can the authors rule out the possibility that the cellular reaction to depletion of H1 is a stress response of the cell that then results in modification/expression of a different gene set that then triggers reactivation? While H1 is ChIP-ed to the EBV genome under the conditions presented, that does not negate the possibility that host factors are also affected and may promote reactivation otherwise.

>Response: While we cannot rule out this point, we wish to emphasize that enforced H1 expression is sufficient to maintain EBV latency upon SFPQ knockout. We performed this experiment to build on the observation that SFPQ KO strongly diminished H1 expression, which we now observe in both EBV+ and EBV- cells. We added a discussion of the possibility that depletion of H1 itself alters gene expression to our discussion (p. 17-18).

- It is not clear how the authors resolve a model where H1 is loaded onto EBV genomes but core histones are not. As described, core histones make up the nucleosome and H1 linker histones bind to the DNA at the entry/exit sites of the nucleosome. That means H1 binds the nucleosome, and in fact, was described to be unable to bind the nucleosome without linker DNA. The affinity of H1 for DNA without histones on it is not discussed here, but suggested in the model where H1 promotes latency, and therefore compaction of the EBV genomes, without the presence of core histones/nucleosomes.

>Response: We apologize for any confusion caused by our original model. We did not intend to imply that the linker histones were loaded onto EBV genomes independently of nucleosomes. In fact, we stated in multiple places in the text that linker H1 functions in the context of core nucleosomes. Rather than wishing to state that H1 acts independently of histones, we agree with the reviewer that H1 loads onto chromatinized EBV genomes and functions in the context of core histones as a linker histone. The name linker histone refers to the fact that H1 is deposited in between nucleosomal DNA. We omitted the nucleosomes from our original final model in an effort to simplify the model and make it less visually busy. However, the reviewer has raised an important point that our initial omission of nucleosomes in the model may imply a model that we did not wish to convey. To prevent any misinterpretation, we revised our model to include both nucleosomes and linker histones H1.2 and H1.4 (Fig. 6d). We also explicitly stated in the figure legend for the ChIP workflow (Fig. 5c) that nucleosomes are not shown for clarity to prevent any misinterpretation of that figure.

Reviewer #2 (Remarks to the Author):

In this manuscript, Murray-Nerger et al. identify a novel role for splicing factor proline and glutamine rich (SFPQ) in Epstein-Barr virus (EBV) latency. The authors previously performed a genome-wide CRISPR/Cas9 loss-of-function screen in Burkitt lymphoma EBV+ B-cells to identify factors that regulate EBV latency (Mol Cell (2020) 78(4):653-669.e8. <https://doi.org/10.1016/j.molcel.2020.03.025>). In this submitted manuscript, the Authors characterized a role for SFPQ, a top hit from the CRISPR/Cas9 screen, in EBV latency. Using independent sgRNAs targeting SFPQ, the authors identify a role for SFPQ in EBV latency maintenance in multiple B- cell and epithelial cell lines. Moreover, the Authors show that SFPQ regulates latency across EBV types (I and II) and for another gammaherpesvirus, KSHV. SFPQ represses EBV lytic replication at the level of IE gene expression through a mechanism that is distinct from that of Myc, another top hit from the CRISPR/Cas9 screen. RNAseq analysis highlighted a role for SFPQ in the regulation of histone gene expression, particularly for linker histones H1.2 and H1.4. In absence of SFPQ, linker histones levels were depleted, H1 occupancy on the EBV genome decreased,

and EBV lytic replication initiated. sgRNA depletion of H1.2 and H1.4 induced EBV lytic replication, whereas enforced H1.2 and H1.4 expression prevented EBV lytic reactivation in absence of SFPQ. The Authors identify that SFPQ regulation of EBV latency requires its DNA-binding-domain, but neither RNA recognition motif. The levels of SFPQ remain stable following EBV lytic reactivation, although SFPQ progressively redistributes into nuclear puncta, which correlates with reduced H1 expression. This manuscript is clearly written, the data is clearly presented, convincing, and largely supports the Authors' conclusions. The submitted work highlights several novel findings, including previously unidentified roles for SFPQ and H1 in EBV latency.

Specific comments:

The relationships between SFPQ, H1, and EBV latency maintenance are convincingly characterized and nicely presented. However, it is not clear whether SFPQ directly or indirectly regulates H1 expression or association with EBV genomes. The Authors present and suggest a model in which SFPQ regulates H1 expression, likely at the H1 transcriptional level due to the requirement for the SFPQ DBD. However, given that H1.2 and H1.4 are DNA-replication-dependent histones within the HIST1 histone cluster, it is not clear how SFPQ regulation of H1 expression to maintain EBV latency conforms with the cell-cycle dependent expression of these histones. Experiments to show that SFPQ associates with H1.2 or H1.4 promoters would strengthen the Author's model that SFPQ supports H1 transcription.

>Response: We agree with the reviewer that demonstrating SFPQ association with the H1.2 promoter is important for strengthening the conclusion that SFPQ regulates H1 expression. We now present ChIP-qPCR analysis, employing both HA-tagged full length (FL) and the DNA binding domain deletion mutant (Δ DBD) SFPQ constructs, to characterize SFPQ association with the H1.2 promoter. These data demonstrate that SFPQ occupies the H1.2 promoter in a DBD-dependent manner. We added these data as new figure panels Fig. 4h and Supplementary Fig. 5g and revised the text on p. 11-12. As discussed above, we now also present data in multiple EBV-negative cell models, including B cells and oral keratinocytes, that loss of SFPQ results in histone H1 depletion.

The SFPQ DBD is necessary for SFPQ to maintain EBV latency. The Authors propose that this is likely due to regulation or support of H1 transcription. Is it possible that SFPQ directly associates with latent EBV genomes to prevent lytic reactivation? Evaluation of whether or not SFPQ associates with latent EBV genomes would further inform the Authors model of SFPQ-mediated maintenance of EBV latency.

>Response: We wish to emphasize that enforced H1 expression was sufficient to maintain EBV latency upon SFPQ CRISPR editing, suggesting that SFPQ support of H1 expression, rather than additional roles on the episome, are critical for maintenance of latency I. However, based on the reviewer's suggestion, we performed ChIP-qPCR analysis, employing both HA-tagged full length (FL) and DNA binding domain deletion mutant (Δ DBD) SFPQ constructs. SFPQ occupied multiple EBV genomic sites, including the *BZLF1* promoter, both *oriLyt* enhancers, and the C promoter (Cp) in a SFPQ DBD-dependent manner (Supplementary Fig. 5h-k and p.

11-12). It is possible that SFPQ plays additional roles in regulation of EBV gene expression, but its support of H1 expression appears to be the critical factor for its role in suppression of lytic reactivation.

Depletion of SFPQ results in decreased H1.2 and H1.4 levels, which is associated with the loss of H1.2 and H1.4 at EBV genomic elements. Is the loss of H1 at these elements an active process or consequent to a global reduction in H1 levels. Is H1 similarly depleted on cellular promoters? Is it possible that SFPQ may facilitate or specifically promote stable association of H1 with EBV latent chromatin?

>Response: Thanks for these points. Based on the reviewer's suggestion, we performed H1.2 ChIP-qPCR analysis to interrogate H1 occupancy of two representative host promoters. The *GAPDH* promoter is constitutively active, and we therefore anticipated low H1.2 levels at this site. In contrast, the *MYO-D* promoter is heterochromatinized in B cells, and we anticipated higher H1.2 association with this site. We employed ChIP-qPCR to analyze H1 levels in latency vs. upon EBV lytic reactivation to assess the change in H1.2 occupancy at these sites. We observed higher association of H1.2 with the *MYO-D* promoter than the *GAPDH* promoter in latent cells, as anticipated. Upon lytic reactivation, H1.2 occupancy decreased at both promoters as early as 6 hours post lytic reactivation. We conclude that H1 loss at the EBV genomic regions is most likely due to a global reduction in H1 levels, rather than a specific active process of H1 removal from the EBV genome or specific promotion by SFPQ of H1 association with the EBV genome. These data are now included in Supplementary Fig. 6g and discussed in the text on p. 13-14. While SFPQ may facilitate or promote H1 association with chromatin, enforced H1 expression was sufficient to maintain EBV latency even in cells depleted for SFPQ.

Minor comments:

In some instances, terminology used is not correct. For example, lines 58/59 "Early genes drive lytic replication of EBV genomes...". Lines 296/297 "an EBV immediate early or early gene may therefore target SFPQ...". "Gene" should be "protein". The Authors should ensure that the terminology used is appropriate.

>Response: We thank the reviewer for these points. We adjusted the text accordingly to clarify this terminology.

Supplementary Figure 1 caption does not match the data presented in the Figure. The text describing Supplementary Figure 3 describes IFIT1 and IRF7 expression in P3HR1 and MUTU I cells (lines 200-201), yet the Figure presents IRF7 expression only in P3HR1 cells. The Authors should ensure that text and accompanying figure captions appropriately describe what is presented in the Figure.

>Response: Thanks for catching this. We revised Supplementary Fig. 1 to match the legend. We also clarified the text (p. 10) to state which cell lines expressed both IFIT1 and IRF7 (P3HR-1) and which show expression of IFIT1 only (MUTU I).

In some instances, information for figures is differently presented. For example, some immunoblots have densitometry quantification (such as Figure 4 b, d; Figure 5 b) whereas others do not (such as Figure 4 g; Figure 5 a). The Authors should ensure that data is presented consistently.

>Response: Thank you for this point. We added densitometry for the original blots that did not have it (Fig. 1d, 1h, 2a, 2d, 2f, 4g, 5a, and 5b and Supplementary Fig. 1f and 6d) as well to the new blots (Supplementary Fig. 5c, 5d, 6b, 8c, 8e, and 8f). All the densitometry quantification is consistent with the findings discussed in the paper.

Reviewer #3 (Remarks to the Author):

This manuscript reports a role for the host SFPQ protein in maintaining EBV latency. The Gewurz lab previously identified SFPQ as a strong hit for genes that help suppress EBV replication in the EBV+ P3HR1 Burkitt lymphoma cell line. Here, they report follow-up experiments, demonstrating its role in multiple EBV+ lymphoma and carcinoma cell lines, and dissecting the mechanism by which SFPQ suppresses EBV replication. They provide evidence that SFPQ acts independently (or possibly downstream) of c-myc - which they had previously reported contributed to maintenance of latency. To further define the mechanism, they perform a CRISPR knockout of SFPQ in P3HR1 cells and find levels of many interferon regulated mRNAs are increased and Histone H1 mRNAs are decreased. They further demonstrate that these Histone H1 proteins bind to regulatory loci on the EBV genome and, most significantly, H1.2 and H1.4 redistribution is temporally associated with lytic reactivation in P3HR1 cells.

The experiments in the manuscript are of high quality and well controlled. Understanding the mechanisms by which EBV remains in a latent state is of fundamental importance, but also has significant implications for translational efforts to treat EBV+ cancers with "lytic induction therapy." Understanding these mechanisms may improve treatment efficacy and help overcome resistance. Additional experiments are not required to convince me that SFPQ plays a role, via Histone H1 in maintaining EBV latency in various EBV+ tumors. However, there is nothing in this manuscript that convinces me that SFPQ is important for establishment of latency upon initial EBV infection (implied in several places, most notably lines 311-327). The experiments presented here rely almost exclusively on a single BL cell line, P3HR1, although Fig 1D does establish the importance of SFPQ in multiple other EBV+ BLs and carcinomas. What is not established is whether SFPQ is important for EBV latency in normal (non-cancer) cells. EBV+ cancers are under strong immune selection pressure against expression of EBV lytic genes. As the authors point out, B cell lymphomas frequently have mutations in multiple histone H1 genes (lines 337-338). It should not be assumed that SFPQ's ability to regulate histone H1 genes to enforce EBV latency is the normal situation and did not arise from similar selection pressure. This remains an interesting unanswered question.

>Response: Thank you for the excellent summary and for these points. The reviewer raises two interesting points regarding: (1) is there a SFPQ role in EBV latency establishment and 2) does

SFPQ support EBV latency in non-cancer cells and/or is it an immunoevasion adaptation that arises from immune pressure to silence EBV antigens in EBV+ cancers.

With regards to the first point, we agree that the establishment versus the maintenance of EBV latency involves a distinct set of reactions. With establishment of latency, there is a need to chromatinize incoming naked DNA genomes, to circularize them, to epigenetically program them, and to orchestrate a genetically defined set of viral genome programs that drive B cell growth and that culminate in the latency III program (it is not currently possible to model latency II or latency I in primary B-cells). We have therefore modified the text, including original lines 311-327, to acknowledge this important point, such that we do not imply that SFPQ has a role in the establishment of latency (p. 18).

Nonetheless, we did expend considerable effort trying to delve into this question. We were limited by the absence of a tractable model system in which to test for key SFPQ roles in latency establishment. A single publication (Akidil et al., *Plos Pathogens*, 2021 PMID: 33857265) reported the use of Cas9 editing in primary B cells concurrent with EBV infection. Even though DNA editing can take place close to the time of infection, decay of the target mRNA and protein can take considerably longer, particularly in non-dividing primary cells (EBV triggers the onset of mitosis at ~day 3 post-infection). Furthermore, in our hands, the published approach yielded SFPQ editing only in a subset of cells, and we had no way to select for this cell population. We could achieve SFPQ knockdown, but not at the levels reached in our Burkitt and gastric and nasopharyngeal carcinoma models. Further complicating matters, primary B cells are short lived in the absence of stimulation, making it difficult to wait for the decay of SFPQ protein following CRISPR editing and prior to EBV infection. Thus, we were unable to address this question in primary cells.

We therefore used two EBV superinfection models to test SFPQ KO effects on EBV gene expression in newly infected cells. First, we used hTERT-immortalized normal oral keratinocytes (NOK), which were originally obtained from gingival tissue removed with a dental extraction. We derived control vs SFPQ edited cell pools and then superinfected with the Akata EBV strain, which encodes a GFP transgene expressed in latency and in the lytic cycle. At 48 h post infection, we observed lower levels of BZLF1 in the SFPQ KO than in control cells and did not observe BMRF1 or EBNA2 in either (EBNA2 is not typically expressed in epithelial cells). Likewise, we superinfected EBV-negative Ramos B cells with GFP+ Akata. Both control and SFPQ edited Ramos cell pools supported GFP expression over the first five days post infection. Since both latent and lytic proteins are expressed at early stages of B cell infection (Mrozek-Gorska et al., *PNAS* 2019 PMID: 31341086), we tested SFPQ KO effects on both classes of EBV proteins. SFPQ KO decreased levels of BZLF1 and BMRF1 at 48 and 120 hours post infection, but increased EBNA2 levels at 48 h post infection. We incorporated these data into Supplementary Fig. 8 and the text on p. 15-16, 18-19. These data suggest SFPQ may not be required to repress lytic genes during the earliest phases of latency establishment. However, it is quite plausible that this may result from deregulation of the EBV genome folding upon infection of SFPQ KO cells. For instance, the EBV genomic 3D configuration changes markedly from linear, dsDNA genomes in newly infected cells to circularized genomes, to folded genomes that adopt distinct configurations between latency states and upon lytic reactivation. Chromatin based epigenetic states are believed to drive these transitions, and H1 supports higher order DNA

structure. Thus, in the context of nascent infection, the absence of SFPQ and lower H1 levels may yield EBV genomes that do not fold properly and therefore are not competent to induce lytic replication.

With regards to the second point, we agree that it would be ideal to include experiments in EBV+ primary cells. Unfortunately, this would require the isolation and manipulation of memory B cell reservoir cells, which are exceedingly rare – they represent 1 in $10^5 - 10^6$ of peripheral blood B cells. There is no known marker to identify these cells while alive; their identification requires them to be fixed and then analyzed for EBER expression. We and the EBV field have therefore relied on transformed cell models to characterize host factors that maintain EBV latency. To address this important point, we now present data in two EBV-uninfected models to validate that SFPQ is important for histone H1 expression. These cells have not undergone immune selective pressure to minimize EBV lytic gene expression. We now present data that CRISPR SFPQ editing reduces H1 expression in non-transformed normal oral keratinocytes (hTERT-immortalized), originally obtained from gingival tissue (Piboonniyom et al., *Cancer Res.* 2003 PMID 12543805). Likewise, we demonstrate that SFPQ editing depletes H1 expression in Ramos cells, which are an EBV-negative lymphoma cell line.

Finally, while the reviewer stated that most of the experiments that we performed were conducted in the P3HR-1 cell line, we would like to emphasize that we conducted most experiments in both P3HR-1 and MUTU I cells and confirmed key results in two gastric carcinoma cell lines, a nasopharyngeal carcinoma cell line, and EBV/KSHV co-infected primary effusion lymphoma cells.

REVIEWERS' COMMENTS

Reviewer #1 (Remarks to the Author):

The revised manuscript presented by Murray-Nerger et al is significantly improved. I commend the authors on taking the suggestions from reviewers to heart, which has led to a more thorough and comprehensive manuscript. All of my concerns have been addressed.

Reviewer #2 (Remarks to the Author):

This manuscript is a revised version of the manuscript in which Murray-Nerger et al describe roles for SFPQ and H1 in the maintenance of EBV latency. In this revised manuscript, Murray-Nerger et al have added figure panels and altered the text and previous figures to address reviewers' concerns. In the revised manuscript, the authors now also show that the SFPQ DNA-binding domain (DBD) mediates association with the EBV (BZLF and Cp promoters, OriLyt enhancers) and cellular (H1.2 promoter) genomes. Furthermore, they establish a role for SFPQ to regulate H1 expression regardless of cellular EBV infection status. In the absence of SFPQ, H1.2 is globally depleted such that H1 association with the EBV genome and evaluated cellular promoters is reduced.

This revised manuscript is clearly written and the data convincing and well presented. The results presented herein address important outstanding questions pertaining to EBV chromatin and latency maintenance that will be of interest to the field.

Very minor specific comments are below.

Supplemental figure 3H and the associated legend present and describe IFIT1 expression in MUTU1 EBV- and MUTU1 EBV+ cells. However, line 212 (pg 10) indicates that IRF7 is evaluated in MUTU1 cells.

It is not clear whether experiments to evaluate the role of SFPQ DBD for EBV or cellular promoter association (Figure 4H, supplemental Figure 5 G-K) were performed in cells that also expressed SFPQ-targeting shRNA. Given that SFPQ normally functions as a dimer, the presence of endogenous full-length SFPQ may confound ChIP analyses to evaluate mutant SFPQ-association with the selected promoters.

It would be helpful for the reader if figures 1F, 2C, and supplemental figure 1A indicated if the copies of EBV genome evaluated were intracellular or extracellular on the figure panel itself (as for supplemental figure 2A).

Reviewer #3 (Remarks to the Author):

This is a resubmission of manuscript that reported a role for the host SFPQ protein in maintaining EBV latency. I reviewed the manuscript previously and felt that the experiment were of high quality and the results would be impactful to the field in terms of our understanding of basic mechanisms of EBV biology and for their translation potential by informing lytic induction strategies. I raised two main issues. First, this was a study of latency maintenance and claims should not extend to latency establishment. They have discussed this point in their response, conceded the main point, and made appropriate edits to the text. The other issue was the heavy reliance on transformed cells. They make the point that my assertion that "experiments presented here rely almost exclusively on a single BL cell line P3HR1" is an overstatement. Perhaps, but it depends on your interpretation of the adverb "almost exclusively." More substantively, they now demonstrate in two tert-immortalized oral keratinocyte lines and the EBV negative Ramos line that SFPQ editing reduce H1 expression. These experiments satisfy the second concern I previously raised. I support publishing this manuscript in its current form.

We thank the reviewers for their careful consideration of our manuscript. We have addressed the remaining textual and figure labeling concerns expressed by reviewer 2, which we feel enhance the clarity of our manuscript. Our point-by-point responses are marked with “>**Response**”.

REVIEWERS' COMMENTS

Reviewer #1 (Remarks to the Author):

The revised manuscript presented by Murray-Nerger et al is significantly improved. I commend the authors on taking the suggestions from reviewers to heart, which has led to a more thorough and comprehensive manuscript. All of my concerns have been addressed.

>Response: We thank the reviewer for their feedback and are happy that all of their concerns have now been addressed.

Reviewer #2 (Remarks to the Author):

This manuscript is a revised version of the manuscript in which Murray-Nerger et al describe roles for SFPQ and H1 in the maintenance of EBV latency. In this revised manuscript, Murray-Nerger et al have added figure panels and altered the text and previous figures to address reviewers’ concerns. In the revised manuscript, the authors now also show that the SFPQ DNA-binding domain (DBD) mediates association with the EBV (BZLF and Cp promoters, OriLyt enhancers) and cellular (H1.2 promoter) genomes. Furthermore, they establish a role for SFPQ to regulate H1 expression regardless of cellular EBV infection status. In the absence of SFPQ, H1.2 is globally depleted such that H1 association with the EBV genome and evaluated cellular promoters is reduced. This revised manuscript is clearly written and the data convincing and well presented. The results presented herein address important outstanding questions pertaining to EBV chromatin and latency maintenance that will be of interest to the field.

Very minor specific comments are below.

Supplemental figure 3H and the associated legend present and describe IFIT1 expression in MUTU1 EBV- and MUTU1 EBV+ cells. However, line 212 (pg 10) indicates that IRF7 is evaluated in MUTU1 cells.

>Response: We appreciate the reviewer catching this error and have now corrected the text so that the correct protein (IFIT1) is referenced in line 212 (p. 10).

It is not clear whether experiments to evaluate the role of SFPQ DBD for EBV or cellular promoter association (Figure 4H, supplemental Figure 5 G-K) were performed in cells that

also expressed SFPQ-targeting shRNA. Given that SFPQ normally functions as a dimer, the presence of endogenous full-length SFPQ may confound ChIP analyses to evaluate mutant SFPQ-association with the selected promoters.

>Response: We thank the reviewer for this point. We have now revised the text (line 246-247, p. 11) to clarify that the ChIP was performed on HA-tagged full length or Δ DBD SFPQ. We have carefully reviewed our manuscript to ensure that all instances of employed CRISPR-mediated SFPQ knockout are explicitly stated. Therefore, it should be clear that endogenous SFPQ was not knocked out in this case. The reviewer is correct that SFPQ functions as a dimer. However, even if endogenous SFPQ is still present, we would imagine that, if the DBD is required for association, a dimer of endogenous SFPQ- Δ DBD would have a weaker association with DNA than endogenous-endogenous or endogenous-FL SFPQ dimers. The persistence of endogenous SFPQ may account for why we observe a reduction, but not complete elimination, of association of SFPQ with the H1.2 promoter upon expression of the Δ DBD mutant.

It would be helpful for the reader if figures 1F, 2C, and supplemental figure 1A indicated if the copies of EBV genome evaluated were intracellular or extracellular on the figure panel itself (as for supplemental figure 2A).

>Response: We agree with the reviewer and have now added these labels for increased clarity.

Reviewer #3 (Remarks to the Author):

This is a resubmission of manuscript that reported a role for the host SFPQ protein in maintaining EBV latency. I reviewed the manuscript previously and felt that the experiment were of high quality and the results would be impactful to the field in terms of our understanding of basic mechanisms of EBV biology and for their translation potential by informing lytic induction strategies. I raised two main issues. First, this was a study of latency maintenance and claims should not extend to latency establishment. They have discussed this point in their response, conceded the main point, and made appropriate edits to the text. The other issue was the heavy reliance on transformed cells. They make the point that my assertion that “experiments presented here rely almost exclusively on a single BL cell line P3HR1” is an overstatement. Perhaps, but it depends on your interpretation of the adverb “almost exclusively.” More substantively, they now demonstrate in two tert-immortalized oral keratinocyte lines and the EBV negative Ramos line that SFPQ editing reduce H1 expression. These experiments satisfy the second concern I previously raised. I support publishing this manuscript in its current form.

>Response: We thank the reviewer for their feedback and are happy the revision of the text and inclusion of additional experiments have satisfied their concerns.